# THE DEBATE ON RLVR REASONING CAPABILITY BOUNDARY: SHRINKAGE, EXPANSION, OR BOTH? A TWO-STAGE DYNAMIC VIEW

## ABSTRACT

The ongoing debate on whether reinforcement learning with verifiable rewards (RLVR) expands or shrinks the reasoning capabilities of large language models (LLMs) is still not fully resolved. Some studies contend that RLVR mainly improves sampling efficiency but at the expense of diversity and exploratory capacity, resulting in *capability boundary shrinkage*. In contrast, others demonstrate that prolonged training can lead to the emergence of novel reasoning strategies, suggesting *capability boundary expansion*. To reconcile these contradictory findings, we theoretically and empirically show that ***both*** perspectives are partially valid—each aligning with a separate phase in an inherent two-stage probability mass dynamic: (1) *Exploitation* stage: initially, the model primarily samples explored high-reward and low-reward tokens, while rarely selecting the potentially optimal token. Positive advantage estimates increase the probability of high-reward tokens and decrease those of low-reward tokens, yet the optimal token's probability remains largely unchanged during this stage. (2) *Exploration* stage: as training advances, the growth rate of previously acquired high-reward tokens slows as their probabilities approach saturation. When a potentially optimal token—now receiving positive advantage estimates—is occasionally sampled, its probability increases, while those of the originally high-reward tokens decrease. This dynamic suggests that over-exploitation during the exploitation stage may lead to capability boundary shrinkage, whereas prolonged training into the exploration stage can promote an expansion of the reasoning capability boundary. Building upon our insights, we revisit the potential of only using relative negative gradients for prolonging training, providing a theoretical and empirical foundation for the development of more advanced reasoning capabilities.

## 1 INTRODUCTION

Reinforcement learning with verifiable rewards (RLVR) has become a key paradigm for substantially enhancing the reasoning abilities of large language models (LLMs), as exemplified by advanced models such as OpenAI's O1 and O3 [33, 46] and DeepSeek-R1 [22]. By optimizing pre-trained or chain-of-thought (CoT) [65] fine-tuned models through verifiable reward signals, RLVR enables LLMs to excel in complex logical tasks such as mathematics [42, 81, 82] and programming [37, 41].

Despite empirical successes, a fundamental question is still hotly debated: *does RLVR genuinely expand the reasoning capabilities of base models beyond their original boundaries?* Current evidence is sharply divided. (1) One line of research [80, 85, 13, 25, 44, 55, 21] argues for **capability boundary shrinkage**, contending that while RLVR improves sampling efficiency, it fails to produce genuinely novel reasoning strategies and may even induce a progressive narrowing of reasoning capabilities during training. Empirical evidence from Yue et al. [80] shows that although RLVR-trained models perform better under small-$k$ sampling (e.g., $k = 1$), base models achieve higher Pass@$k$ when $k$ is large. Similarly, Cui et al. [12] document a sharp entropy collapse during training, resulting in overly deterministic behavior [83] and reduced exploratory effectiveness. (2) In contrast, another body of work [39, 66, 38, 68, 79, 63] provides evidence supporting **capability boundary expansion**. Liu et al. [39] attribute previous evidence of capability boundary shrinkage to the premature termination of RL training, which disrupts learning before novel reasoning capabilities can

fully develop. Through prolonged training, they further demonstrate that RLVR can explore and populate new regions of solution space over time. Meanwhile, Wu et al. [68] experimentally show that RLVR can occasionally expand empirical support, producing novel correct solutions beyond the original reach of the base model.

*The debate between these two lines of evidence centers on empirical results; however, the underlying mechanisms responsible for these contradictory findings remain unclear.* To elucidate the mechanisms, we focus on the evolution of the policy model's probability mass distribution—termed the *probability mass dynamics*. As a conceptual starting point, consider that the search tree [80, 90, 24] for any given prompt is built through iterative sampling from the policy. This tree grows exponentially at a rate of $\mathcal{O}(V^T)$, where $V$ denotes the vocabulary space (token set) size and $T$ the maximum generation length. Crucially, policy updates can be viewed as a dynamic reallocation of probability mass across the search tree, thereby shaping the reasoning capability boundary.

Through an integrated theoretical and empirical analysis (Section 3), **we demonstrate that both lines of evidence hold validity to some extent**—each corresponding to a distinct stage within a two-stage dynamic of probability mass. Specifically, since the logit for token $v$ is directly tied to its policy probability—a larger (smaller) logit results in a higher (lower) probability—we analyze the policy gradient of the training objective and derive a bidirectional update rule for the logits (i.e., the pre-Softmax values; Lemma 1). According to this rule, updates to the logits depend on both the advantage estimate $\hat{A}$ and the current policy distribution $\pi$. Under practical optimization settings such as GRPO [56] (where multiple responses are sampled per prompt), Theorem 1 establishes that the expected logit update for token $v$ is proportional to $\pi(v)\left[(1 - \pi(v))\hat{A}(v) - \sum_{u\neq v} \pi(u)\hat{A}(u)\right]$.

From this view, the overall dynamic appears to unfold in two distinct stages. (1) **Exploitation stage**: initially, the model predominantly samples the already-explored high-reward token and the low-reward token, while the potentially optimal token is selected only infrequently. Driven by positive advantage estimates, the probability of the high-reward token increases, whereas that of the low-reward token decreases. However, the probability of the potentially optimal token remains largely unchanged throughout this stage. This behavior suggests that *over-exploitation during this stage may result in a shrinkage of the capability boundary*. (2) **Exploration stage**: as training progresses, the growth rate of the high-reward token previously explored slows as its probability approaches near saturation ($1-\pi \to 0$). When the potentially optimal token—now associated with positive advantage estimates—is occasionally sampled, its probability increases, while that of the formerly high-reward token declines. A key characteristic of this dynamic is the transition of the relative negative sample: from the initially low-reward token to the high-reward token. This implies that with prolonged training, gradient updates can be progressively redirected toward tokens with low initial probability but high potential, once high-probability tokens have stabilized, ultimately *expanding the reasoning capability boundary*. We illustrate these theoretical insights with a toy example (Section 3.2).

Building on our theoretical and experimental insights, a direct way to expand the reasoning capability boundary and mitigate shrinkage is to prolong training while concentrating policy probability updates exclusively on optimizing relative negative samples (denoted -N, Section 4.1) throughout the learning process. Empirical investigations (Section 4) of our strategy—implemented in widely adopted algorithms (e.g. GRPO, GSPO [89]) on benchmark datasets and open-source LLMs verify that GRPO-N (GSPO-N) achieves competitive and stable performance improvements while largely preserving the base model's diversity, demonstrating the potential for prolonged training. Notably, analysis of the training process reveals instances where incorrect code is initially generated but is later refined and corrected through iterative reflection. Unlike GRPO, which reinforces the entire trajectory—including error-prone steps—GRPO-N effectively prevents such reinforcement.

◇ **Main contributions**. Briefly, this study unveils the underlying mechanisms responsible for the heated debate (boundary shrinkage or expansion) in RLVR from both theoretical and practical perspectives. We emphasize the essential role of fine-grained probability mass allocation and establish a theoretical and empirical basis for understanding the impact of RLVR on reasoning capabilities.

## 1.1 MORE RELATED WORKS

Broadly speaking, our work builds upon lines of research in reinforcement learning for LLM reasoning, LLM learning dynamics, and gradient analysis in preference optimization. A comprehensive review of related work is included in Appendix A due to page constraints.

## 2   PRELIMINARIES AND BACKGROUND

In this section, we describe the core components of our study by reviewing some basic notations.

**RLVR**. Reinforcement learning with verifiable rewards (RLVR) is a paradigm for improving models on tasks with objectively verifiable outcomes. In this formulation, an autoregressive language model is treated as a policy $\pi_\theta$ (parameter $\theta$). For a given query $\mathbf{x}$ from a prompt set $\mathcal{D}$, the probability of generating a response $\mathbf{y}$ is defined as $\pi_\theta(\mathbf{y} \mid \mathbf{x}) = \prod_{t=1}^{|\mathbf{y}|} \pi_\theta(y_t \mid \mathbf{x}, \mathbf{y}_{<t})$. A deterministic reward function $r$ assigns a scalar value indicating the correctness of the full response $\mathbf{y}$ to the prompt $\mathbf{x}$. Each token in $\mathbf{y}$ receives the same reward (1 only if the final answer is correct, and 0 otherwise). The objective is to minimize the loss: $\mathcal{L}_{\text{RLVR}}(\theta) = -\mathbb{E}_{\mathbf{x}\sim\mathcal{D}, \, \mathbf{y}\sim\pi_\theta(\cdot|\mathbf{x})} [r(\mathbf{x}, \mathbf{y})]$, where $r(\mathbf{x}, \mathbf{y}) \in [0, 1]$.

**A unified framework for policy gradient optimization**. Building on the work of [56, 36, 59], we consider a unified objective $\mathcal{J}$ that establishes connections among various optimization methods:

$$\mathcal{J}_{\text{RLVR}}(\theta) = \mathbb{E}_{\mathbf{x}\sim\mathcal{D}, \, \mathbf{y}\sim\pi_{\theta_{\text{old}}}(\cdot|\mathbf{x})} \left[ \frac{1}{|\mathbf{y}|} \sum_{t=1}^{|\mathbf{y}|} \min\left( w_t(\theta)\hat{A}_t, \text{clip}\left( w_t(\theta), 1 - \epsilon, 1 + \epsilon \right)\hat{A}_t \right) \right], \quad (1)$$

where $\epsilon$ is a clipping hyperparameter, $\text{clip}(\cdot)$ is the clipping operation, and the the importance ratio of the token $y_t$ is defined as $w_t(\theta) = \frac{\pi_\theta(y_t|\mathbf{x},\mathbf{y}_{<t})}{\pi_{\theta_{\text{old}}}(y_t|\mathbf{x},\mathbf{y}_{<t})}$ (the current policy $\pi_\theta$ and the old policy $\pi_{\theta_{\text{old}}}$). $\hat{A}_t$ is the advantage of current token and is implemented differently across optimization methods:

• **PPO** (Proximal Policy Optimization [54, 48]). $\hat{A}_t$ is computed by applying Generalized Advantage Estimation (GAE) [53], based on the value model. This incurs considerable computational and memory overhead, and its effectiveness critically depends on the reliability of its value estimation.

• **GRPO** (Group Relative Policy Optimization [56]). To reduce variance, GRPO and its variants (e.g., DAPO [78] & Dr.GRPO [40]) eliminate reliance on a value model by using Monte Carlo estimates to compute the relative advantage across a group of responses $\{\mathbf{y}_i\}_{i=1}^{G} \sim \pi_{\theta_{\text{old}}}$ to the same query (where $G$ is the group size and all token in $\mathbf{y}_i$ share the same relative advantage):

$$w_{i,t}(\theta) = \frac{\pi_\theta(y_{i,t} \mid \mathbf{x}, \mathbf{y}_{i,<t})}{\pi_{\theta_{\text{old}}}(y_{i,t} \mid \mathbf{x}, \mathbf{y}_{i,<t})}, \quad \hat{A}_{i,t} = \hat{A}_i = \frac{r(\mathbf{x}, \mathbf{y}_i) - \text{mean}\left( \{r(\mathbf{x}, \mathbf{y}_i)\}_{i=1}^{G} \right)}{\text{std}\left( \{r(\mathbf{x}, \mathbf{y}_i)\}_{i=1}^{G} \right)}.$$

• **GSPO** (Group Sequence Policy Optimization [89]). Given that the token-level importance ratio $w_{i,t}$ in GRPO does not align with sequence-level rewards, GSPO introduces a sequence-level importance ratio $w_i$ based on sequence likelihood [88]:

$$w_i(\theta) = \left( \frac{\pi_\theta(\mathbf{y}_i \mid \mathbf{x})}{\pi_{\theta_{\text{old}}}(\mathbf{y}_i \mid \mathbf{x})} \right)^{\frac{1}{|\mathbf{y}_i|}} = \exp\left( \frac{1}{|\mathbf{y}_i|} \sum_{t=1}^{|\mathbf{y}_i|} \log \frac{\pi_\theta(y_{i,t} \mid \mathbf{x}, \mathbf{y}_{i,<t})}{\pi_{\theta_{\text{old}}}(y_{i,t} \mid \mathbf{x}, \mathbf{y}_{i,<t})} \right).$$

To better understand the model's learning dynamics under this binary outcome reward setting, we omit the regularization components[1] (e.g., KL term & clipping operation). That is, the policy gradient $\nabla_\theta \mathcal{J}_{\text{RLVR}}(\theta)$ can be simplified to $\mathbb{E}\left[ \frac{1}{|\mathbf{y}|} \sum_{t=1}^{|\mathbf{y}|} w_t(\theta)\hat{A}_t \nabla_\theta \log \pi_\theta(y_t \mid \mathbf{x}, \mathbf{y}_{<t}) \right]$ with respect to $\theta$. Specifically, taking GRPO as an example (Appendix B.1 for derivation):

$$\nabla_\theta \mathcal{J}_{\text{GRPO}}(\theta) = \mathbb{E}_{\mathbf{x},\{\mathbf{y}_i\}_{i=1}^{G}} \left[ \frac{1}{G} \sum_{i=1}^{G} \frac{1}{|\mathbf{y}_i|} \sum_{t=1}^{|\mathbf{y}_i|} \underbrace{w_{i,t}(\theta)\hat{A}_{i,t}}_{\text{coefficient}} \nabla_\theta \log \pi_\theta(y_{i,t} \mid \mathbf{x}, \mathbf{y}_{i,<t}) \right]. \quad (2)$$

**Remark 1.** *Intuitively, if we set $\hat{A}_{i,t} = 1$ and $w_{i,t} = 1$ while all $\mathbf{y}_i$ are correct responses, then Eq.(1) essentially performs maximum likelihood estimation, i.e., supervised fine-tuning (SFT). Furthermore, Eq.(2) indicates that the scalar $w_{i,t}\hat{A}_{i,t}$ can be interpreted as a weighting coefficient that adjusts the log-likelihood term. This implies that RLVR methods can be viewed as a form of reweighted SFT, where correct responses and incorrect responses contribute positive and negative gradients, respectively [56, 10, 91, 15, 1, 8]. When $\hat{A}_{i,t}$ is calculated from a comparison of average rewards across groups (e.g., GRPO), the resulting gradient is named the relative policy gradient.*

---

[1]Regularization components are widely regarded as mechanisms for ensuring training stability. Moreover, studies [31, 10] indicate that omitting them does not impair performance when others are properly tuned.

## 3 PROBABILITY MASS DYNAMICS

As described above, we begin by considering a standard task that involves generating a reasoning sequence. In this setting, the model learns a policy $\pi_\theta(\mathbf{y} \mid \mathbf{x}) = \prod_{t=1}^{T} \pi_\theta(y_t \mid \mathbf{x}, \mathbf{y}_{<t}) \in \mathbb{R}^{V \times T}$ to map an input $\mathbf{x}$ to a sequence of predictions $\mathbf{y} = \{y_1, \ldots, y_T\}$, where $\mathbf{y} \in \mathcal{V}^T$, $\mathcal{V}$ is the vocabulary space of size $V$, and $T$ denotes the maximum generation length. Conceptually, the reasoning process can be regarded as a tree search [80, 90, 24]. A search tree is constructed for a given problem by iteratively sampling from the policy model. This process leads to exponential growth in the tree size, $\mathcal{O}(V^T)$, reflecting an open-ended and combinatorially infinite reasoning space [47].

Crucially, policy updates can be viewed as dynamically reallocating probability mass over the search tree, thereby shaping the boundary of reasoning capability. Here, we specifically focus on the evolution of the policy model's probability distribution—referred to as probability mass dynamics.

$\diamond$ Learning dynamics offer critical insights into the key challenges and counterintuitive behaviors of deep learning [52], with early explanations pointing to network "stiffness" [20] or "local elasticity" [26, 17]. To track the evolution of the probability distribution, we monitor the logits $\mathbf{z}^\theta \in \mathbb{R}^{V \times T}$ and the log probabilities $\log \pi_\theta(\mathbf{y} \mid \mathbf{x})$, where $\pi_\theta$ is derived from $\mathbf{z}^\theta$ via a column-wise Softmax $\pi_\theta(\cdot \mid \mathbf{x}, \mathbf{y}_{<t}) = \text{Softmax}(\mathbf{z}^\theta(\mathbf{x}, \mathbf{y}_{<t}))$. The **probability mass dynamics** are then defined as:

$$\Delta \mathbf{z}^l(\mathbf{x}) \triangleq \mathbf{z}^{\theta^{l+1}}(\mathbf{x}) - \mathbf{z}^{\theta^l}(\mathbf{x}), \tag{3}$$

$$\Delta \log \pi^l(\mathbf{y} \mid \mathbf{x}) \triangleq \log \pi_{\theta^{l+1}}(\mathbf{y} \mid \mathbf{x}) - \log \pi_{\theta^l}(\mathbf{y} \mid \mathbf{x}), \tag{4}$$

where the model's parameter $\theta$ is updated from step $l$ to $l+1$ by performing one policy gradient update on the sample data $(\mathbf{x}, \mathbf{y})$. For simplicity, we primarily analyze the case where $T = 1$ (i.e., $\mathbf{y} \in \mathcal{V}$), meaning $\Delta \mathbf{z}^l \in \mathbb{R}^{V \times 1}$ and its dimension aligns with the size of the model's vocabulary. Notably, a larger (smaller) logit results in a higher (lower) probability. For $T > 1$, the updates can be computed separately; therefore, we can calculate the the distinct $T$ updates and stack them together.

### 3.1 A TWO-STAGE DYNAMIC: EXPLOITATION AND EXPLORATION

Given the monotonicity of the Softmax function, the main text focuses mainly on characterizing the changes of $\mathbf{z}$. Analysis of the updates to $\log \pi_\theta$ with respect to $\theta$ is provided in the Appendix B.3.

**Lemma 1** (Logits Update for Softmax Parameterization). *Consider a policy parameterized by a Softmax function over logits $\mathbf{z}(\mathbf{x}) := \mathbf{z} = [z_1, \cdots, z_V]^T$, such that the probability of action (or token) $v$ is given by $\pi(v) := \pi(v \mid \mathbf{x}) = \text{Softmax}(\mathbf{z})_v = \exp(z_v)/\sum_{v'}^{V} \exp(z_{v'})$. Reviewing Eq.(2), if the currently sampled action is $v$, let the policy gradient estimate be $\nabla_{\mathbf{z}} \mathcal{J} \approx \hat{A}(v) \nabla_{\mathbf{z}} \log \pi(v)$. For a learning rate $\eta$, the update rule for the logits at time step $l$ is (Appendix B.2 for derivation):*

- *For the sampled action $v$:*

$$z_v^{l+1} \leftarrow z_v^l + \eta \cdot \hat{A}(v) \cdot \left(1 - \pi^l(v)\right), \quad \Delta z_v^l = \eta \cdot \hat{A}(v) \cdot \left(1 - \pi^l(v)\right),$$

- *For all other actions $u \neq v$:*

$$z_u^{l+1} \leftarrow z_u^l + \eta \cdot \hat{A}(v) \cdot \left(-\pi^l(u)\right), \quad \Delta z_u^l = \eta \cdot \hat{A}(v) \cdot \left(-\pi^l(u)\right).$$

**Remark 2** (Bidirectional Update Rule). *The update to the logit $\mathbf{z} \in \mathbb{R}^{V \times 1}$ depends on both the advantage estimate $\hat{A}$ and the current policy distribution $\pi$. Specifically, Let $v$ denote the currently sampled action. (1) when $\hat{A}(v) > 0$: $z_v$ increases by $\eta \hat{A}(v)(1 - \pi(v))$ while $z_u$ ($u \neq v$) decreases by $\eta \hat{A}(v) \pi(u)$; (2) when $\hat{A}(v) < 0$: $z_v$ decreases by $\eta |\hat{A}(v)|(1 - \pi(v))$ while $z_u$ ($u \neq v$) increases by $\eta |\hat{A}(v)| \pi(u)$. The normalization property of Softmax ensures that when $\hat{A}(v) > 0$, the update increases $\pi(v)$ while decreasing $\pi(u)$ for all $u \neq v$, including other advantageous actions. In contrast, when $\hat{A}(v) < 0$, the update increases the probabilities of other actions proportionally to their current policy values. The update may reallocate probability mass toward other potentially advantageous actions that were previously under-sampled.*

The practical update in group policy optimization (e.g. GRPO, DAPO, GSPO, REINFORCE++ [30], GPG [10], GPO [77]), which employs Monte Carlo sampling, arises from the collective effect of a group of responses, thus motivating our analysis of the **expected logits update**.

**Theorem 1** (The Expected Logits Update). *Under the conditions stated in Lemma 1, we assume[2] that $\mathbf{x} \sim \mathcal{D}$ is i.i.d. and $\{u_i\}_{i=1}^{G}$ are randomly sampled from $\pi(\cdot \mid \mathbf{x})$, the expected group relative policy gradient $\nabla_{\mathbf{z}} \mathcal{J} \in \mathbb{R}^{V \times 1}$ is $\mathbb{E}_{\mathbf{x} \sim \mathcal{D}, \{u_i\}_{i=1}^{G} \sim \pi(\cdot|\mathbf{x})} \left[ \frac{1}{G} \sum_{i=1}^{G} \hat{A}(u_i) \nabla_{\mathbf{z}} \log \pi(u_i) \right]$. Then the expected logits update is (proof in Appendix B.4):*

$$\mathbb{E}(\Delta z_v^l) = \eta \cdot \pi^l(v) \left[ (1 - \pi^l(v))\hat{A}(v) - \sum_{u \neq v} \pi^l(u)\hat{A}(u) \right].$$

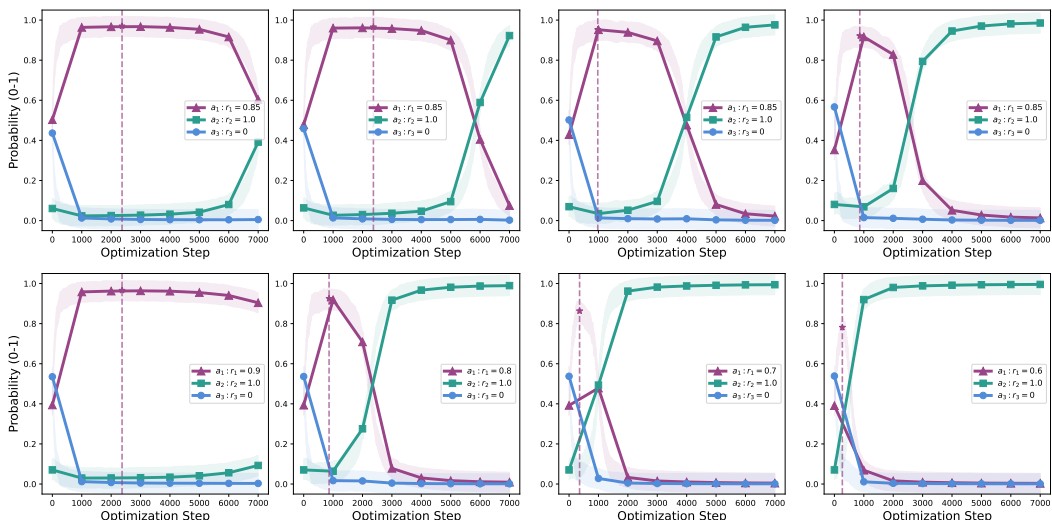

Figure 1: The probability mass dynamics of policy optimization across varying action rewards $r$ and initial policy probabilities $\pi$. Each sub-figure corresponds only to the different rewards and probabilities. The first row compares the impact of different initial policy probabilities under identical rewards, while the second row compares the effect of varying rewards given the same initial policy.

**Remark 3** (A Two-Stage Dynamic of Exploitation and Exploration). *Theorem 1 establishes that the magnitude of the expected logit update $\Delta z_v^l$ is explicitly governed by $\pi^l(v)$. Although the Softmax function guarantees strictly positive probabilities for all actions, a significant number of these actions lie within the extreme tail of the probability distribution. As a result, under finite-sample training conditions, such actions exert negligible influence on parameter updates and are effectively omitted during optimization (that is, $\pi^l(v) \to 0$ leads to $\Delta z_v^l \to 0$). Interestingly, the overall dynamic appears to unfold in two distinct stages. (1) **Exploitation stage, corresponding to capability boundary shrinkage**: initially, the model mainly samples already-explored high-reward and low-reward tokens, rarely selecting the potentially optimal one. Driven by positive advantage estimates, the probability of the high-reward token increases while the low-reward token decreases. The potentially optimal token's probability remains largely unchanged (or may even decrease, Remark 2), suggesting that over-exploitation in this stage may cause capability boundary shrinkage. (2) **Exploration stage, corresponding to capability boundary expansion**: As training continues, the growth of the previously dominant high-reward token slows as it approaches saturation ($1 - \pi \to 0$). When the potentially optimal token—now receiving positive advantage signals—is occasionally sampled, its probability rises, while that of the former high-reward token decreases. A key feature of this stage is the shift in relative negative sampling: from the initial low-reward token to the once high-reward token. This implies that through prolonged training, gradient updates can be shifted toward tokens with low initial probability but high reward, once high-probability tokens have stabilized. For instance, under the Pass@$k$ metric, raising the probability of at least one correct action above $1/k$ corresponds to an expansion of the reasoning capability boundary.*

---

[2]Without loss of generality, we approximate importance ratio $w \approx 1$, as regularization components such as the KL penalty and clipping operation are applied in practical training.

## 3.2 Demonstration with A Toy Example

Next, to more clearly demonstrate the theoretically predicted two-stage dynamic, we validate the above analysis of probability mass dynamics using a simple toy setting, and subsequently review several widely adopted RLVR tricks and more than three actions case in Appendix C.4.

◇ **Starting with a toy setting**. To better track the probability mass dynamics, we analyze the scenario in a clean and simplified setting, assuming the entire action space consists of only three actions[3]: $a_1$, with $r(a_1) > 0$, which has been explored; $a_2$, with $r(a_2) > 0$, which remains unexplored; and $a_3$, with $r(a_3) = 0$, which has been explored. Let the initial logits be denoted as $\mathbf{z} = [z(a_1), z(a_2), z(a_3)]^T$, and the policy as $\pi(a_i) = \exp(z(a_i))/\sum_{j \neq i} \exp(z(a_j)), \forall i \in [1, 2, 3]$. Here, we perform $G$ action samplings, estimate the relative advantages via $\hat{A}(a_i) = r(a_i) - \text{mean}(\{r(a_j)\}_{j=1}^G)$, and subsequently update the logits $\mathbf{z}$ using the policy gradient update rule given in Theorem 1. As stated in Remark 3, we discuss the following scenario[4]: RLVR reinforces high-probability yet suboptimal actions while overlooking potentially optimal correct actions that initially have low probability, thereby leading to over-exploitation behavior. That is, $r(a_1) < r(a_2)$, while initially $\pi(a_1) > \pi(a_2)$. For implementation details, see Algorithm 1.

**Results for demonstration**. We sample actions at each optimization step (with $G = 2, \eta = 0.1$) and analyze the probability mass dynamics (a larger $G$ leads to more stable optimization and does not affect our main findings and conclusions). Figure 1 clearly illustrates the dynamics of the probability mass of policy optimization across different rewards $r$ & initial policy probabilities $\pi$, which aligns with the theoretical analysis in Section 3.1. That is, $\mathbb{E}(\Delta z(a_i)) = \eta\pi(a_i)\left[(1 - \pi(a_i))\hat{A}(a_i) - \sum_{j \neq i}^3 \pi(a_j)\hat{A}(a_j)\right]$. More specifically, the overall dynamics can be divided into two stages: (1) Initially, $\pi(a_1)$ and $\pi(a_3)$ are relatively large while $\pi(a_2)$ is comparatively small. Since actions $a_1$ and $a_3$ are predominantly sampled, and given that $\hat{A}(a_1) > 0$ and $\hat{A}(a_3) < 0$, $\pi(a_1)$ increases while $\pi(a_3)$ decreases. Meanwhile, $\pi(a_2)$ remains almost unchanged. (2) As $\pi(a_1)$ increases, the gradient term $1 - \pi(a_1)$ gradually approaches zero, causing the growth of $\pi(a_1)$ to stabilize. If training continues beyond this point, when action $a_2$ is sampled with $\hat{A}(a_2) > 0$ and $\hat{A}(a_1) < 0$, $\pi(a_1)$ will decrease while $\pi(a_2)$ increases. Note that throughout the optimization process, the relative negative actions change (initially $a_3$ and later $a_1$).

**Remark 4.** *From the two-stage dynamics, (1) it can be observed that although the relative policy gradient method does exhibit the phenomenon of capability boundary shrinkage. However, prolonging the duration of the training may result in further gradient updates being applied to low-probability action sequences once the high-probability ones have reached convergence. This is precisely why the research represented by Cui et al. [12] employs entropy control mechanisms to extend the duration of training. (2) More interestingly, the relative policy gradient may undergo changes during the training process: $\pi(a_1)$ first increases and then decreases. Therefore, simply using the momentum of policy gradients from the early stages of updates—as in methods like AAPO [71]—to enhance policy optimization is suboptimal. In contrast, approaches such as ProRL [39, 38] periodically **reset the reference policy and optimizer states** during training.*

## 4 How to Prolong Training: Revisiting the Role of Relative Negative Gradients

Thus far, we have established the imperative of avoiding over-sharpening in the policy distribution—which induces over-exploitation and entropy collapse—and of enabling sustained training. Liu et al. [39] identify a fundamental limitation across existing studies [80, 13, 85]: RL training is frequently terminated prematurely after only a few hundred steps, hindering the models' ability to fully explore and acquire novel reasoning capabilities. Their conclusions align closely with our findings. Therefore, enhancing training stability and facilitating extended training durations constitute promising directions for future research.

---

[3]In Section 2, the reward $r$ is sparse (1 or 0). However, due to the presence of factors such as the importance ratio $w$, distinctions arise among positive rewards ($wr$). For brevity of analysis, we ignore the standard deviation $\text{std}(\{r(a_j)\}_{j=1}^G)$ because it does not affect the sign (positive or negative nature) of $\hat{A}(a_i)$.

[4]Otherwise, we can proceed with normal optimization to increase the probability of the optimal action.

In group policy optimization (e.g. GRPO, GSPO), the policy $\pi_\theta(\mathbf{y} \mid \mathbf{x}) = \prod_{t=1}^{|\mathbf{y}|} \pi_\theta(y_t \mid \mathbf{x}, \mathbf{y}_{<t})$ learned by the model is inherently complex. Returning to Section 3, policy updates can be interpreted as dynamically redistributing probability mass across the search tree, which has a size of $\mathcal{O}(V^T)$. To unlock the model's capacity for genuinely novel reasoning, we call for research into strategies that more effectively allocate probability mass. Based on the probability mass dynamics established in Lemma 1 and Theorem 1, optimizing relative negative advantage actions implicitly increases the probability of other actions. A straightforward strategy is to allocate policy probability mass exclusively through relative negative gradients within the overall dynamics. In this part, we will revisit the role of only using relative negative gradients in prolonging training.

## 4.1 EXPERIMENTAL SETUP

We choose `Qwen2.5-Math-7B` [74] and `Llama-3.2-3B-Instruct` [60] as our base models for investigation, which align with our hardware resource. **For RLVR algorithms**, we evaluate the standard approach alongside a variant that employs exclusively relative negative gradients[5] (denoted as **-N**). This comparison includes widely-used methods such as GRPO [22, 56] and GSPO [89]. Moreover, we use the verl framework [57] to train the models and the detailed hyperparameter settings of training and evaluation can be found in Appendix C.2. **For the datasets**, we employ the training set of MATH [28], which comprises 7,500 problems, for model training (with prompt batch size of 1,024). Performance is evaluated on widely-used reasoning benchmarks, (1) in-domain (ID) tasks: the test sets of MATH, AIME 2024, AIME 2025, and AMC 2023. (2) out-of-domain (OOD) tasks: ARC-c [11] (open-domain reasoning), MMLU-Pro [64] (academic reasoning).

Specifically, we adopt Pass@$k$ as our primary evaluation metric, which measures whether a model can successfully solve a problem within $k$ attempts. This metric has been widely used to mitigate the unreliability of greedy decoding-based accuracy estimates [29] and to better assess the true capability boundaries of models [7, 9, 80, 91]. The unbiased estimator first generates the $n$ responses for per question $\mathbf{x}$ ($n \geq k$), counts the number of correct responses $c$, then computes the metric as:

$$\text{Pass@}k = \mathbb{E}_{\mathbf{x} \sim \mathcal{D}} \left[ 1 - \frac{\binom{n-c}{k}}{\binom{n}{k}} \right].$$

## 4.2 TRAINING DYNAMICS AND EVALUATION RESULTS

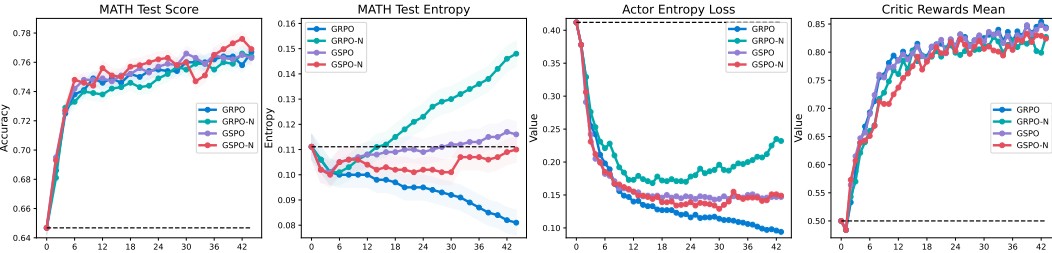

Figure 2: Comparison of the training dynamics of GRPO, GRPO-N, GSPO, and GSPO-N on the MATH benchmark across training steps, using the `Qwen2.5-Math-7B` model with a prompt batch size of 1,024. **Left Part:** (Left) the greedy decoding accuracy on the MATH test set and (Center Left) the model's entropy on the MATH test set. **Right Part:** (Center Right) the actor entropy loss and (Right) critic rewards mean during training. GRPO causes the entropy of the base model to collapse over the course of training, suggesting a loss of exploratory capability. In contrast, GRPO-N, GSPO, and GSPO-N all exhibit a pattern where entropy initially decreases and then increases. Notably, the entropy of GRPO-N significantly surpasses that of the base model. All algorithms achieve competitive performance in both greedy decoding accuracy and critic rewards mean.

**Training dynamics.** We characterize the training dynamics by monitoring the greedy decoding accuracy and entropy on a held-out MATH test set over the course of training (Figure 2 for

---

[5]The relative advantage $\hat{A}_{i,t}$ is computed over a group of responses. When $\hat{A}_{i,t} < 0$, the term $w_{i,t}(\theta)\hat{A}_{i,t}\nabla_\theta \log \pi_\theta(y_{i,t} \mid \mathbf{x}, \mathbf{y}_{i,<t})$ is referred to as a relative negative gradient. See Appendix B.5 for the **details of relative negative gradients**.

Qwen2.5-Math-7B, Figure 4 for Llama-3.2-3B-Instruct), together with the actor entropy loss and critic rewards mean during training. As illustrated, GRPO, GRPO-N, GSPO, and GSPO consistently achieve competitive performance in both greedy decoding accuracy and critic rewards mean. Notably, GRPO leads to a rapid and substantial decline in entropy on the MATH test set. In contrast, GRPO-N, GSPO, and GSPO-N all show an initial decrease in entropy, followed by a consistent increase. Importantly, the entropy on the held-out test set under GRPO-N significantly exceeds that of the base model. This divergence indicates that the standard GRPO may limit output diversity and exploratory capability (see Table 1), both methods that apply sequence-level importance ratio clipping directly (GSPO and GSPO-N) and those that utilize only relative negative gradients (GRPO-N) help mitigate overconfidence in previously sampled responses. Of particular significance, prior study [12] suggests that policy performance comes at the cost of policy entropy, and is therefore bottlenecked by its exhaustion. Therefore, the model optimized by GRPO-N may be a good baseline and maintain the base model's diversity for prolonging training[6].

Table 1: Evaluation results of Qwen2.5-Math-7B on in-domain tasks (AMC 2023, AIME 2024, and AIME 2025) and out-of-domain tasks (ARC-c and MMLU-Pro). For each $k$, **bold** and underlined numbers indicate the best and second-best results, respectively.

| Algorithm | Pass@$k$ | | | | | | | | |
|---|---|---|---|---|---|---|---|---|---|
| $k$ | 1 | 2 | 4 | 8 | 16 | 32 | 64 | 128 | 256 |
| **AMC 2023** | | | | | | | | | |
| Base Model | 40.4 | 55.6 | 69.1 | 79.4 | 85.9 | 89.5 | 92.1 | 94.6 | 97.5 |
| GRPO | 60.4 | 69.9 | 77.4 | 82.9 | 86.7 | 89.4 | 91.7 | 94.7 | 97.5 |
| GRPO-N | 59.2 | 68.7 | 76.3 | 82.7 | 87.6 | **92.3** | **96.3** | **99.1** | **100.0** |
| GSPO | 61.1 | 70.5 | 78.0 | 83.9 | 88.1 | 91.6 | 94.4 | 96.2 | 97.5 |
| GSPO-N | **61.5** | **71.2** | **78.5** | **84.1** | **88.4** | 91.8 | 94.8 | 97.4 | 100.0 |
| **AIME 2024** | | | | | | | | | |
| Base Model | 13.6 | 21.8 | 30.5 | 37.5 | 43.5 | 49.7 | 55.8 | 61.4 | 66.7 |
| GRPO | 22.6 | 31.5 | 39.5 | 46.2 | 51.9 | 57.3 | 62.9 | 68.9 | 73.3 |
| GRPO-N | 23.6 | 33.4 | 41.8 | 47.5 | 51.9 | 56.7 | 61.8 | 67.3 | 73.3 |
| GSPO | **25.3** | **34.7** | **42.4** | 48.3 | 53.6 | 58.7 | 63.6 | 68.1 | 73.3 |
| GSPO-N | 23.3 | 31.1 | 42.1 | **48.8** | **54.3** | **59.4** | **64.4** | **69.6** | **73.3** |
| **AIME 2025** | | | | | | | | | |
| Base Model | 6.4 | 10.2 | 14.5 | 18.9 | 23.6 | 28.1 | 32.5 | 38.3 | 46.7 |
| GRPO | 9.2 | 13.4 | 17.9 | 22.4 | 26.4 | 29.9 | 33.9 | 39.1 | 46.7 |
| GRPO-N | 9.5 | 14.2 | 19.1 | 23.8 | 28.7 | 34.2 | **41.6** | **52.5** | **66.7** |
| GSPO | 9.6 | 14.3 | 19.2 | 23.9 | 28.8 | 34.2 | 40.9 | 49.5 | 60.0 |
| GSPO-N | **10.2** | **14.7** | **19.6** | **24.9** | **29.9** | **35.0** | 40.9 | 47.3 | 53.3 |
| **ARC-c** | | | | | | | | | |
| Base Model | 35.4 | 54.9 | 73.7 | 86.5 | 93.6 | 96.9 | 98.2 | 99.2 | 100.0 |
| GRPO | 62.3 | 77.4 | 86.6 | 91.6 | 94.2 | 96.3 | 98.2 | 99.5 | 100.0 |
| GRPO-N | 61.7 | **78.1** | **88.5** | **94.3** | **97.7** | **99.5** | **99.9** | **100.0** | **100.0** |
| GSPO | 59.9 | 74.1 | 83.2 | 89.1 | 93.2 | 95.7 | 96.7 | 96.9 | 96.9 |
| GSPO-N | **63.9** | 77.9 | 86.5 | 91.1 | 93.7 | 95.5 | 96.5 | 96.9 | 96.9 |
| **MMLU-Pro** | | | | | | | | | |
| Base Model | 28.1 | 41.4 | 55.1 | 67.6 | 78.1 | **85.9** | **91.6** | **96.2** | **100.0** |
| GRPO | 40.1 | 52.0 | **62.4** | **70.8** | 76.9 | 80.6 | 83.7 | 87.3 | 90.6 |
| GRPO-N | 38.5 | 49.9 | 60.7 | 70.1 | **78.1** | 84.3 | 89.0 | 93.7 | 100.0 |
| GSPO | 40.0 | 50.6 | 60.6 | 69.2 | 75.5 | 79.5 | 82.4 | 85.9 | 90.6 |
| GSPO-N | **41.6** | **52.3** | 61.8 | 69.7 | 75.9 | 79.9 | 83.6 | 88.1 | 93.8 |

**Performance on ID&OOD tasks**. As shown in Table 1, for model with strong prior (e.g., Qwen models), both GRPO-N and GSPO-N consistently achieve a favorable trade-off across various values of $k$ on both ID tasks (e.g., AMC 2023, AIME 2024, and AIME 2025) and OOD tasks (such as ARC-c and MMLU-Pro). In particular, (1) GSPO-N matches the best Pass@1 performance on AMC 2023,

---

[6] For model with weak prior, we provide the training dynamics and evaluation results of LLama-3.2-3B-Instruct in Appendix C.3. The performance ceiling is related to the base model, yet the key finding remains consistent across different models.

AIME 2025, ARC-c and MMLU-Pro. (2) GRPO-N and GSPO-N reliably improve the reasoning performance of the base model on ID tasks for every value of $k$. (3) For OOD tasks, GRPO-N (GSPO-N) achieves higher Pass@$k$ scores than GRPO (GSPO) across all $k$ values, demonstrating stable performance improvements while largely preserving the diversity of the base model.

In this subsection, we demonstrate that while all algorithms achieve competitive performance, they exhibit distinct behaviors at the entropy level. This observation motivates further analysis of the **differences among various correct responses** in the next part.

### 4.3 ANALYSIS OF DIFFERENT CORRECT RESPONSES

**Question:** Consider the set of complex numbers $z$ satisfying $|1 + z + z^2| = 4$. The maximum value of the imaginary part of $z$ can be written in the form $\frac{\sqrt{m}}{n}$, where $m$ and $n$ are relatively prime positive integers. What is $m + n$?

**Ground-truth Answer: 21**

Let's break down the problem… **1.** Let $z = x + yi$, where $x$ and $y$ are real numbers. **2.** Substitute $z$ and $z^2$ into the given equation, and squaring both sides: $(1 + x + x^2 - y^2)^2 + (y + 2xy)^2 = 16$. **3.** We need to find the maximum value of $y$ that satisfies the above equation. Let's use Python and SymPy to solve this. <…**Python code**…> < …**TypeError**… > **4.** The error occurs because…… revised code: <…**Python code**…> < …**TypeError** … > **5.** < **Reproduce revised code behavior**…> **6.** The numerical approach can be computationally intensive and may not be the most efficient way to solve this problem. Instead, let's use a more analytical approach…… **7.** By analyzing the equation, we can see that the maximum value of $y$ occurs when the real part of the expression inside the magnitude is minimized and the imaginary part is maximized. This happens when $x = -1/2$ which simplifies the equation to:

$$\left(1 - \frac{1}{2} + \left(-\frac{1}{2}\right)^2 - y^2\right)^2 + \left(y + 2\left(-\frac{1}{2}\right)y\right)^2 = 16$$

**8.** Taking the square root of both sides, we get: $\left|\frac{3}{4} - y^2\right| = 4$. Since $y^2$ cannot be negative. Solving the second case: $\frac{3}{4} - y^2 = -4$. $y = \pm\frac{\sqrt{19}}{2}$. The values $m$ and $n$ are 19 and 2, respectively…... Therefore, $m + n = 19 + 2 = 21$. The final answer is:

\[
\boxed{21}
\]

**GRPO**

Let's break down the problem… **1.** Let $z = x + yi$, where $x$ and $y$ are real numbers. **2.** Square both sides to remove the square root: $(1 + x + x^2 - y^2)^2 + (y + 2xy)^2 = 16$. **3.** Let's proceed with a numerical approach using Python to find the maximum value of $y$. Let's write the Python code to perform this search. <…**Python code**…><…Output…> **4.** The numerical method suggests that the maximum value of $y$ is approximately $-0.03$, which is very close to 0. This indicates that the maximum imaginary part of $z$ is likely 0, but let's verify this by checking the exact form of the solution. **5.** Let's consider the equation again: $(1 + x + x^2 - y^2)^2 + (y + 2xy)^2 = 16$. **6.** We can see that the maximum imaginary part occurs when $x = -1/2$. Substituting $x = -1/2$ into the equation, we get:

$$\left(1 - \frac{1}{2} + \frac{1}{4} - y^2\right)^2 + (y - y)^2 = \left(\frac{3}{4} - y^2\right)^2 = 16$$

**7.** Solving for $y$, we get: $y^2 = \frac{3}{4}x \pm 4$. Since $y^2$ must be non-negative, we take the positive root: $y = \pm\frac{\sqrt{19}}{2}$. Therefore, $m = 19$ and $n = 2$, so $m + n = 21$. Thus, the final answer is:

\[
\boxed{21}
\]

**GRPO-N**

Figure 3: A comparison of the correct responses of GRPO and GRPO-N (a test case from AMC 2023). The key reasoning steps are presented here, see Appendix C.7 for full procedure.

**Case study**. We present the reasoning processes of GRPO and GRPO-N, as illustrated in Figure 3. Although various responses can yield correct answers through reasoning, their underlying reasoning behaviors differ markedly. Specifically, GRPO tends to repeat similar errors, such as persistently generating code with TypeErrors, and fails to rectify them. In contrast, GRPO-N produces fewer erroneous codes than GRPO. We further examined the responses generated during the training process and found instances where incorrect code was initially produced but later reflected upon and corrected to form the final correct response. GRPO assigns higher probability to the entire trajectory (i.e., the behavior of generating incorrect code is reinforced—a tendency that may significantly affect the model's self-correction ability), whereas GRPO-N mitigates this issue. This necessitates an inquiry into achieving finer-grained control of probability assignments.

## 5 CONCLUSION AND LIMITATION

**Conclusion**. Based on a two-stage dynamic view of probability mass allocation, this study resolves the ongoing debate on whether RLVR shrinks or expands LLM reasoning capabilities. We show that initial training favors exploitation, potentially narrowing capability boundaries, while prolonged training encourages exploration, enabling genuine expansion. Theoretically and empirically, we demonstrate that both phenomena occur at different phases. Guided by these findings, one can develop new algorithms to foster more advanced reasoning capabilities.

**Limitation**. However, further studies are required on (i) how to design efficient algorithms for fine-grained probability mass allocation; (ii) what kind of base models are more conducive to capability boundary expansion during the RL stage; and (iii) where the ceiling of boundary exploration lies. We leave these questions for our future work.

# 6   MORE DISCUSSION

**Theoretical Contribution**. **A good theory should not only explain observations but also inform design**. The paper's main contribution lies in its two-stage framework for interpreting reasoning dynamics in RLVR, not in introducing a strong algorithm. Guided by these findings, one can develop new algorithms to foster more advanced reasoning capabilities. The GRPO-N strategy serves as a straightforward implementation validating the guiding principles of our theory. Additionally, Appendix C.5 provides a method for finer-grained control of probability assignments, inspired by our theoretical analysis.

**Negative Reward Gradients & Negative Advantage Gradients**. Here, we further clarify the methodological and conceptual differences between our work and NSR [91]. (1) As documented in Appendix D.3 of the NSR paper, normalization is disabled in their implementation, and the advantage is defined as the raw reward (i.e., +1 for PSR and -1 for NSR). Consequently, NSR is more precisely described as method using **negative reward gradients**, which does not account for relative changes over the course of training. (2) In contrast, GRPO-N uses **negative advantage gradients** ($\frac{r(\mathbf{x},\mathbf{y}_i)-\text{mean}\big(\{r(\mathbf{x},\mathbf{y}_i)\}_{i=1}^{G}\big)}{\text{std}\big(\{r(\mathbf{x},\mathbf{y}_i)\}_{i=1}^{G}\big)}$). As discussed in our theoretical analysis (Thereom 1), a key feature of the two-stage dynamic is the shift in relative negative samples: from initially low-reward tokens (Figure 1 $a_3$) to previously high-reward tokens (Figure 1 $a_1$). Consequently, here, positive reward samples still serve the function of computing relative advantages by providing learning signals.

**Rationale for Omitting Regularization (KL, Clipping)**. (1) For theoretical analysis. This simplification was made primarily to highlight the core two-stage dynamics of policy optimization under a simplified setting, allowing us to isolate and analyze the fundamental learning stages. (2) For real training. As we stated in Section 2, we acknowledge that in practical implementations, both clipping and KL regularization play important roles in stabilizing training. However, **such simplifications can be justified by empirical evidence**: prior work [31] has shown that omitting the clipping operation does not lead to performance degradation, while studies [31, 10] have demonstrated that the KL term can be removed when other hyperparameters are appropriately tuned. (3) For the effect of clipping and KL-regularization. Cui et al. [12] report that despite the reference KL achieves stable entropy values, it fails to improve policy and instead leads to a degradation in performance. Yu et al. [78] identify that the upper clip can restrict the exploration of the policy, where making an 'exploitation' token more probable is much easier yet the probability of an unlikely 'exploration' token is too tightly bounded to be uplifted. Therefore, excessive KL-regularization and clipping will lead to limited exploration and premature policy determinization, thereby suppressing the emergence of the second phase in the two-stage dynamics.

**Stability Analysis of GSPO/GSPO-N and GRPO/GRPO-N**. (1) First, we recall the key distinction between GSPO and GRPO, as outlined in Section 2: while the token-level importance ratio $w_{i,t}$ in GRPO may not align well with sequence-level rewards, GSPO introduces a sequence-level importance ratio $w_i$ based on sequence likelihood. (2) Moreover, certain tokens (such as "wait") that frequently appear in both positive and negative rollouts can lead to instability when optimized with token-level importance ratios. As illustrated in Figure 2, this explains why both GSPO and GSPO-N exhibit greater stability compared to GRPO and GRPO-N. (3) For theoretical explanation, from a high-level understanding perspective, you can view the sequence as a holistic action. Therefore, the sequence-level importance ratio $(w_i(\theta))$ mitigates, to some extent, the influence of the initial policy's $\pi(\cdot \mid \mathbf{x})$ value on the expected policy gradient ($\mathbb{E}_{\mathbf{x}\sim\mathcal{D},\{u_i\}_{i=1}^{G}\sim\pi(\cdot|\mathbf{x})}\left[\frac{1}{G}\sum_{i=1}^{G}\hat{A}(u_i)\nabla_{\mathbf{z}}\log\pi(u_i)\right]$).

**Potential Future Directions**. (1) For finer-grained control of probability assignments: Under current outcome-only reward schemes (in GRPO), trajectories with incorrect intermediate python calls can still receive positive reward if the final answer is correct, effectively reinforcing the model to treat such errors as acceptable (Figure 3). Intuitively, we can filter correct responses based on the count of correct Python (or tools) call executions they contain. (2) To promote diversity, we can employ text similarity metrics [16] to filter the samples. This may help mitigate excessive exploration in the first stage and encourages a transition to the second. For instance, in E-GRPO [87], a dense entity-aware reward is formulated based on the number of ground-truth entities identified, which by definition creates a task-dependent reward. In the future, one can focus on exploring task-independent approaches that eliminate the need for explicit step-wise decomposition and scoring.

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

# A    MORE RELATED WORKS

Here, we discuss more related works to supplement the main text.

**Reinforcement Learning for LLM Reasoning**. Large language models (LLMs) are often post-trained using reinforcement learning (RL), both for preference alignment [48, 5] and to improve reasoning capabilities [56, 22]. Inspired by Shao et al. [56], Liu [36] and Swamy et al. [59], this work reformulates methods like SFT, RFT [62], DPO, PPO, and GRPO as maximum likelihood estimation governed by a *Gradient Coefficient*. This coefficient fundamentally operates by amplifying gradients for favored responses and suppressing others, with its magnitude modulating the preference intensity. Thus, the core challenge in policy gradient methods reduces to the accurate estimation of this *Gradient Coefficient* (i.e., the advantage and importance ratio). For instance, AAPO [71] redefines advantage estimation by incorporating advantage momentum. GAPO [34], GVPO [84] and $\Delta L$ Normalization [27] employ gradient normalization to adaptively rescale each objective's gradients, thereby finding a low-variance estimator. Meanwhile, Zhao et al. [86] and Qian et al. [50] utilize a model's own internal confidence measure (or entropy)—termed self-certainty to improve reasoning skills. Additionally, hybrid approaches that integrate RL with SFT on external demonstration data have been actively explored [6, 43, 73, 69, 19]. Despite these empirical advances, the fundamental question of whether RLVR expands [39, 66, 38, 68, 79, 63, 4] or shrinks [80, 85, 58, 13, 25, 44, 55, 21, 14, 45] the reasoning capacities of LLMs remains an open and actively debated issue. *This is precisely what we aim to uncover.*

**LLM Learning Dynamics**. Deep neural networks learn by adjusting their parameters through gradient descent. This process, known as learning dynamics, connects how model predictions change to the gradients from individual training examples. *Learning dynamics prioritizes the analysis of a model's relative training behavior over its convergence, providing a means to assess the quality of individual training samples*. To name a few, Pruthi et al. [49] introduce "TracIn", a metric that measures how much a training example affects a model's predictions, Xia et al. [70] later use it to identify the most influential examples during instruction fine-tuning of LLMs. In a similar vein, Guo et al. [23] propose a method based on the neural tangent kernel (NTK) regime to estimate the relative difficulty among different training samples. Furthermore, Ren & Sutherland [52] highlight a unique "squeezing effect" to explain a previously observed phenomenon in off-policy direct preference optimization (DPO [51]), where running DPO for too long makes even the desired outputs less likely. Since RLVR methods—exemplified by PPO and GRPO—are on-policy and dynamically evolving, we argue that analyzing learning dynamics can naturally offer a novel perspective for understanding the hot debate (capability boundary shrinkage or expansion) in RLVR.

**Gradient Analysis in Preference Optimization**. DPO [51] has proven highly effective, as it relies solely on an offline dataset of paired preference data. However, this reliance on paired data restricts its applicability in settings where only unpaired feedback (e.g., solely positive or negative responses) is available. In response, Abdolmaleki et al. [1] introduce a decoupled approach that independently controls the influence of positive and negative signals, enabling learning even when only a single feedback type is available. Regarding online update methods, RAFT++ [18, 72]—a simple rejection sampling approach utilizing only positively rewarded data—has been shown to deliver performance competitive with GRPO. Conversely, Zhu et al. [91] report the surprising effectiveness of training exclusively on negatively rewarded samples using REINFORCE [67], without reinforcing correct responses. *As we demonstrate in the main text, the set of samples considered "negative" is not static but evolves dynamically throughout optimization*. It imperative to analyze the underlying learning dynamics. In addition, Yang et al. [75] and Chen et al. [8] find that negative responses hold learning value (e.g., self-reflection). However, existing methods overlook this by either discarding them (RFT) or applying uniform penalties (RL), failing to leverage these nuanced signals. There are also some token-level gradient analyses: Yang et al. [76] identify that RL training is skewed by low-probability tokens' excessive gradient magnitudes, impeding the learning from essential high-probability tokens; Deng et al. [15] empirically observe that GRPO can suffer from what we call Lazy Likelihood Displacement: a failure to sufficiently increase, or even a decrease in, the likelihood of correct answers during training. *The above motivates us to analyze the expected update in RLVR, once again emphasizing the essential role of fine-grained probability mass allocation.*

## LLM USAGE

Regarding the use of LLMs, they were employed solely for language polishing purposes and played no role in research ideation, literature retrieval, or any other academically substantive activities.

## B  OMITTED PROOFS AND ADDITIONAL RESULTS

### B.1  PROOF OF EQUATION 2

*Proof.* We begin by reviewing the objective function of GRPO below.

$$\mathcal{J}_{\text{GRPO}}(\theta) = \mathbb{E}_{\mathbf{x}\sim\mathcal{D},\ \{\mathbf{y}_i\}_{i=1}^{G}\sim\pi_{\theta_{\text{old}}}(\cdot|\mathbf{x})}$$
$$\left[\frac{1}{G}\sum_{i=1}^{G}\frac{1}{|\mathbf{y}_i|}\sum_{t=1}^{|\mathbf{y}_i|}\left\{\min\left(w_{i,t}(\theta)\hat{A}_{i,t},\text{clip}\left(w_{i,t}(\theta),1-\epsilon,1+\epsilon\right)\hat{A}_{i,t}\right)\right\}-\beta\mathbb{D}_{\text{KL}}(\pi_\theta\ ||\ \pi_{\text{ref}})\right],$$

where $w_{i,t}(\theta) = \frac{\pi_\theta(y_{i,t}|\mathbf{x},\mathbf{y}_{i,<t})}{\pi_{\theta_{\text{old}}}(y_{i,t}|\mathbf{x},\mathbf{y}_{i,<t})}$, $\mathbb{D}_{\text{KL}}(\pi_\theta\ ||\ \pi_{\text{ref}}) = \frac{\pi_{\text{ref}}(y_{i,t}|\mathbf{x},\mathbf{y}_{i,<t})}{\pi_\theta(y_{i,t}|\mathbf{x},\mathbf{y}_{i,<t})} - \log\frac{\pi_{\text{ref}}(y_{i,t}|\mathbf{x},\mathbf{y}_{i,<t})}{\pi_\theta(y_{i,t}|\mathbf{x},\mathbf{y}_{i,<t})} - 1$, $\beta$ is the coefficient.

To better understand the model's learning dynamics under this binary outcome reward setting, we omit the regularization components (e.g., KL term & clipping operation):

$$\mathcal{J}_{\text{GRPO}}(\theta) = \mathbb{E}_{\mathbf{x}\sim\mathcal{D},\ \{\mathbf{y}_i\}_{i=1}^{G}\sim\pi_{\theta_{\text{old}}}(\cdot|\mathbf{x})}\left[\frac{1}{G}\sum_{i=1}^{G}\frac{1}{|\mathbf{y}_i|}\sum_{t=1}^{|\mathbf{y}_i|}w_{i,t}(\theta)\hat{A}_{i,t}\right],$$

$$\nabla_\theta\mathcal{J}_{\text{GRPO}}(\theta) = \mathbb{E}_{\mathbf{x}\sim\mathcal{D},\ \{\mathbf{y}_i\}_{i=1}^{G}\sim\pi_{\theta_{\text{old}}}(\cdot|\mathbf{x})}\left[\frac{1}{G}\sum_{i=1}^{G}\frac{1}{|\mathbf{y}_i|}\sum_{t=1}^{|\mathbf{y}_i|}\nabla_\theta w_{i,t}(\theta)\hat{A}_{i,t}\right]$$

$$= \mathbb{E}_{\mathbf{x}\sim\mathcal{D},\ \{\mathbf{y}_i\}_{i=1}^{G}\sim\pi_{\theta_{\text{old}}}(\cdot|\mathbf{x})}\left[\frac{1}{G}\sum_{i=1}^{G}\frac{1}{|\mathbf{y}_i|}\sum_{t=1}^{|\mathbf{y}_i|}\frac{\nabla_\theta\pi_\theta(y_{i,t}\mid\mathbf{x},\mathbf{y}_{i,<t})}{\pi_{\theta_{\text{old}}}(y_{i,t}\mid\mathbf{x},\mathbf{y}_{i,<t})}\hat{A}_{i,t}\right]$$

$$= \mathbb{E}_{\mathbf{x}\sim\mathcal{D},\ \{\mathbf{y}_i\}_{i=1}^{G}\sim\pi_{\theta_{\text{old}}}(\cdot|\mathbf{x})}\left[\frac{1}{G}\sum_{i=1}^{G}\frac{1}{|\mathbf{y}_i|}\sum_{t=1}^{|\mathbf{y}_i|}\frac{\pi_\theta(y_{i,t}\mid\mathbf{x},\mathbf{y}_{i,<t})}{\pi_{\theta_{\text{old}}}(y_{i,t}\mid\mathbf{x},\mathbf{y}_{i,<t})}\hat{A}_{i,t}\nabla_\theta\log\pi_\theta(y_{i,t}\mid\mathbf{x},\mathbf{y}_{i,<t})\right]$$

$$= \mathbb{E}_{\mathbf{x}\sim\mathcal{D},\ \{\mathbf{y}_i\}_{i=1}^{G}\sim\pi_{\theta_{\text{old}}}(\cdot|\mathbf{x})}\left[\frac{1}{G}\sum_{i=1}^{G}\frac{1}{|\mathbf{y}_i|}\sum_{t=1}^{|\mathbf{y}_i|}\underbrace{w_{i,t}(\theta)\hat{A}_{i,t}}_{\text{coefficient}}\nabla_\theta\log\pi_\theta(y_{i,t}\mid\mathbf{x},\mathbf{y}_{i,<t})\right].$$

We complete the proof of Equation 2. Notice that $w_{i,t}$ does not affect the sign of $\hat{A}_{i,t}$.

Besides, one can also consider the gradient of the KL term (denote $\pi(y_{i,t}\mid\mathbf{x},\mathbf{y}_{i,<t})$ as $\pi(y_{i,t})$):

$$\nabla_\theta\beta\mathbb{D}_{\text{KL}}(\pi_\theta\ ||\ \pi_{\text{ref}}) = \beta\nabla_\theta\frac{\pi_{\text{ref}}(y_{i,t})}{\pi_\theta(y_{i,t})} - \beta\nabla_\theta\log\frac{\pi_{\text{ref}}(y_{i,t})}{\pi_\theta(y_{i,t})}$$

$$= -\beta\frac{\pi_{\text{ref}}(y_{i,t})}{\pi_\theta^2(y_{i,t})}\nabla_\theta\pi_\theta(y_{i,t}) + \beta\nabla_\theta\log\pi_\theta(y_{i,t})$$

$$= -\left[\beta\frac{\pi_{\text{ref}}(y_{i,t})}{\pi_\theta(y_{i,t})} - \beta\right]\nabla_\theta\log\pi_\theta(y_{i,t}).$$

$\square$

## B.2 PROOF OF LEMMA 1

*Proof.* Re-stating the Lemma 1, the output of a model is the logits $\mathbf{z} = [z_1, ..., z_V]^T$, which corresponds to a finite (size $V$) vocabulary set $\mathcal{V} = \{v_1, ..., v_V\}$. The policy probability of the corresponding action (token) is calculated by: $\pi(v) = \text{Softmax}(\mathbf{z})_v = \exp(z_v) / \sum_{v'}^V \exp(z_{v'})$.

That is, $\log \pi(v) = \log(\exp(z_v)) - \log(\sum_{v'}^V \exp(z_v')) = z_v - \log(\sum_{v'}^V \exp(z_v'))$.

Thus, for the currently sampled token $v$, let $z_v$ be its corresponding logit, we will have:

$$\frac{\partial \log \pi(v)}{\partial z_v} = 1 - \pi(v),$$

for other unsampled tokens $u \neq v$ (with its logit $z_u$):

$$\frac{\partial \log \pi(v)}{\partial z_u} = -\pi(u).$$

Apply those to the gradient $\nabla_{\mathbf{z}} \mathcal{J} = \hat{A}(v) \nabla_{\mathbf{z}} \log \pi(v)$, we complete the proof of Lemma 1. $\square$

## B.3 PROPOSITION 1 AND PROOF

**Proposition 1.** *Let the conditions specified in Lemma 1 hold, and denote $\Delta \mathbf{z}(\mathbf{x}) = [\Delta z_1, ..., \Delta z_V]^T$, the $l$-th step probability mass dynamics decompose as:*

$$\Delta \log \pi^l(\mathbf{y} \mid \mathbf{x}) = \left[I - \mathbf{e}(\pi^l(\mathbf{y} \mid \mathbf{x}))^T\right] \left[(\nabla_\theta \mathbf{z}^{\theta^l}(\mathbf{x}))(\nabla_\theta \mathbf{z}^{\theta^l}(\mathbf{x}))^T\right] \Delta \mathbf{z}^l(\mathbf{x}) + \mathcal{O}(\eta^2 \left\|\nabla_\theta \mathbf{z}^{\theta^l}(\mathbf{x})\right\|^2),$$

*where $I$ is the identity matrix and $\mathbf{e} = [1, 1, ..., 1]^T$, $\left[(\nabla_\theta \mathbf{z}^{\theta^l}(\mathbf{x}))(\nabla_\theta \mathbf{z}^{\theta^l}(\mathbf{x}))^T\right] \in \mathbb{R}^{V \times V}$ is the empirical neural tangent kernel, $\Delta \log \pi^l(\mathbf{y} \mid \mathbf{x}) \in \mathbb{R}^{V \times 1}$. $\Delta \mathbf{z}(\mathbf{x}) = \eta \nabla_{\mathbf{z}} \mathcal{J} \in \mathbb{R}^{V \times 1}$, which mainly determines the **direction and magnitude** of the policy update.*

*Proof.* Recall the log probabilities change in Eq. (4):

$$\Delta \log \pi^l(\mathbf{y} \mid \mathbf{x}) \triangleq \log \pi_{\theta^{l+1}}(\mathbf{y} \mid \mathbf{x}) - \log \pi_{\theta^l}(\mathbf{y} \mid \mathbf{x}) := \log \pi^{l+1}(\mathbf{y} \mid \mathbf{x}) - \log \pi^l(\mathbf{y} \mid \mathbf{x}),$$

and we follow Ren & Sutherland [52] using Taylor expansion to approximate $\log \pi^{l+1}(\mathbf{y} \mid \mathbf{x})$:

$$\log \pi^{l+1}(\mathbf{y} \mid \mathbf{x}) = \log \pi^l(\mathbf{y} \mid \mathbf{x}) + \langle \nabla \log \pi^l(\mathbf{y} \mid \mathbf{x}), \theta^{l+1} - \theta^l \rangle + \mathcal{O}(\left\|\theta^{l+1} - \theta^l\right\|^2).$$

Then, supposing the parameters' are updated by policy gradient, we will have (the model parameters $\theta \in \mathbb{R}^{d \times 1}$):

$$\Delta \log \pi^l(\mathbf{y} \mid \mathbf{x}) = \nabla_\theta \log \pi^l(\mathbf{y} \mid \mathbf{x})(\theta^{l+1} - \theta^l) + \mathcal{O}(\left\|\theta^{l+1} - \theta^l\right\|^2).$$

Next, we use the definition of gradient and the chain rule:

$$\nabla_\theta \log \pi^l(\mathbf{y} \mid \mathbf{x})(\theta^{l+1} - \theta^l) = \left[\nabla_{\mathbf{z}^{\theta^l}} \log \pi^l(\mathbf{y} \mid \mathbf{x})(\nabla_\theta \mathbf{z}^{\theta^l}(\mathbf{x}))\right] [\eta \nabla_{\theta^l} \mathcal{J}]^T$$

$$= \left[\nabla_{\mathbf{z}^{\theta^l}} \log \pi^l(\mathbf{y} \mid \mathbf{x})(\nabla_\theta \mathbf{z}^{\theta^l}(\mathbf{x}))\right] \left[\eta \nabla_{\mathbf{z}^{\theta^l}} \mathcal{J}(\nabla_\theta \mathbf{z}^{\theta^l}(\mathbf{x}))\right]^T$$

$$= \nabla_{\mathbf{z}^{\theta^l}} \log \pi^l(\mathbf{y} \mid \mathbf{x}) \left[(\nabla_\theta \mathbf{z}^{\theta^l}(\mathbf{x}))(\nabla_\theta \mathbf{z}^{\theta^l}(\mathbf{x}))^T\right] (\eta \nabla_{\mathbf{z}^{\theta^l}} \mathcal{J})$$

$$= \left[I - \mathbf{e}(\pi^l(\mathbf{y} \mid \mathbf{x}))^T\right] \left[(\nabla_\theta \mathbf{z}^{\theta^l}(\mathbf{x}))(\nabla_\theta \mathbf{z}^{\theta^l}(\mathbf{x}))^T\right] \Delta \mathbf{z}^l(\mathbf{x}).$$

For the higher-order term:

$$\theta^{l+1} - \theta^l = \eta \nabla_{\theta^l} \mathcal{J} = \eta (\nabla_\theta \mathbf{z}^{\theta^l}(\mathbf{x}))^T \nabla_{\mathbf{z}^{\theta^l}} \mathcal{J},$$

and from the practical application and Lemma 1, the term $\nabla_{\mathbf{z}^{\theta^l}} \mathcal{J}$ is usually bounded, we get:

$$\mathcal{O}(\left\|\theta^{l+1} - \theta^l\right\|^2) = \mathcal{O}(\eta^2 \left\|\nabla_\theta \mathbf{z}^{\theta^l}(\mathbf{x})\right\|^2).$$

We complete the proof. $\left[(\nabla_\theta \mathbf{z}^{\theta^l}(\mathbf{x}))(\nabla_\theta \mathbf{z}^{\theta^l}(\mathbf{x}))^T\right] \in \mathbb{R}^{V \times V}$ denotes the empirical neural tangent kernel (NTK), which remains nearly constant throughout the training process [52, 3, 32]. As a result, $\Delta \mathbf{z}^l(\mathbf{x})$ primarily governs both the direction and magnitude of the policy update. $\square$

## B.4   PROOF OF THEOREM 1

**Theorem 1.** Under the conditions stated in Lemma 1, we assume that $\mathbf{x} \sim \mathcal{D}$ is i.i.d., the expected group relative policy gradient $\nabla_{\mathbf{z}} \mathcal{J} \in \mathbb{R}^{V \times 1}$ is $\mathbb{E}_{\mathbf{x} \sim \mathcal{D}, \{u_i\}_{i=1}^{G} \sim \pi(\cdot|\mathbf{x})} \left[ \frac{1}{G} \sum_{i=1}^{G} \hat{A}(u_i) \nabla_{\mathbf{z}} \log \pi(u_i) \right]$. Then the expected logits update is:

$$\mathbb{E}(\Delta z_v^l) = \eta \cdot \mathbb{E}_{u \sim \pi^l(\cdot|\mathbf{x})} \left[ \hat{A}(u) \nabla_{z_v^l} \log \pi^l(u) \right] = \eta \cdot \pi^l(v) \left[ (1 - \pi^l(v)) \hat{A}(v) - \sum_{u \neq v} \pi^l(u) \hat{A}(u) \right].$$

*Proof.* From Lemma 1, the policy gradient of sampling a token (action) $u$ once from the policy $\pi^l(\cdot \mid \mathbf{x})$ is $\hat{A}(u) \nabla_{\mathbf{z}} \log \pi(u)$. Thus, the expected group relative policy gradient is the following:

$$\nabla_{\mathbf{z}} \mathcal{J} = \mathbb{E}_{\mathbf{x} \sim \mathcal{D}, \{u_i\}_{i=1}^{G} \sim \pi(\cdot|\mathbf{x})} \left[ \frac{1}{G} \sum_{i=1}^{G} \hat{A}(u_i) \nabla_{\mathbf{z}} \log \pi(u_i) \right].$$

Given that $\mathbf{x} \sim \mathcal{D}$ is i.i.d. and $\{u_i\}_{i=1}^{G}$ are randomly sampled from $\pi(\cdot \mid \mathbf{x})$, we derive an unbiased estimator:

$$\nabla_{\mathbf{z}} \mathcal{J} = \mathbb{E}_{u \sim \pi(\cdot|\mathbf{x})} \left[ \hat{A}(u) \nabla_{\mathbf{z}} \log \pi(u) \right] = \sum_u \pi(u) \hat{A}(u) \nabla_{\mathbf{z}} \log \pi(u) \in \mathbb{R}^{V \times 1}.$$

Apply those to Lemma 1, we complete the proof. $\qquad\square$

## B.5   DETAILS OF RELATIVE NEGATIVE GRADIENTS

Referring back to Eq.(1) and Eq.(2), taking GRPO as an example, we obtain the gradient of the objective function in the following form.

$$\nabla_\theta \mathcal{J}_{\text{GRPO}}(\theta) = \mathbb{E}_{\mathbf{x}, \{\mathbf{y}_i\}_{i=1}^{G} \sim \pi_{\theta_{\text{old}}}(\cdot|\mathbf{x})} \left[ \frac{1}{G} \sum_{i=1}^{G} \frac{1}{|\mathbf{y}_i|} \sum_{t=1}^{|\mathbf{y}_i|} \underbrace{w_{i,t}(\theta) \hat{A}_{i,t}}_{\text{coefficient}} \nabla_\theta \log \pi_\theta(y_{i,t} \mid \mathbf{x}, \mathbf{y}_{i,<t}) \right].$$

Since the advantage $\hat{A}_{i,t}$ is estimated from the currently sampled group $i = 1, \cdots, G$ each time, we refer to it as the relative advantage, and correspondingly, this gradient is termed the relative policy gradient. Consequently, for the relative negative gradients, we exclusively utilize gradient information where $\hat{A}_{i,t} < 0$ during the gradient update:

$$\nabla_\theta \mathcal{J}_{\text{GRPO-N}}(\theta) = \mathbb{E}_{\mathbf{x}, \{\mathbf{y}_i\}_{i=1}^{G} \sim \pi_{\theta_{\text{old}}}(\cdot|\mathbf{x})} \left[ \frac{1}{G} \sum_{i=1}^{G} \frac{1}{|\mathbf{y}_i|} \sum_{t=1}^{|\mathbf{y}_i|} \mathbb{I}(\hat{A}_{i,t}) \cdot w_{i,t}(\theta) \hat{A}_{i,t} \nabla_\theta \log \pi_\theta(y_{i,t} \mid \mathbf{x}, \mathbf{y}_{i,<t}) \right],$$

where $\mathbb{I}(\hat{A}_{i,t})$ is an indicator variable that equals 1 if $\hat{A}_{i,t} < 0$, and 0 otherwise.

## C   EXTENSION TO EXPERIMENTS

**Reproducibility statement**. We employed open-source algorithms and data to validate our theoretical analysis, and have reported all hyperparameter settings to facilitate reproducibility.
(1) open-source code: https://github.com/volcengine/verl.
(2) all datasets can be found in: https://huggingface.co/datasets.
(3) toy example details are provided in: Algorithm C.1.

### C.1   ALGORITHM FOR LOGITS UPDATE

---

**Algorithm 1** Logits Update for Softmax Parameterization: A Toy Example

---

**Require:** learning rate $\eta$, number of samples per update $G$, true rewards $r$, optimization steps $N$
Initialize policy parameters (logits) $\mathbf{z}$
**for** $l = 1$ to $N$ **do**
    Compute current policy $\pi \leftarrow \text{Softmax}(\mathbf{z})$
    Sample $G$ actions from policy $\pi$: $\{a_1, a_2, ..., a_G\}$
    Estimate advantage $\hat{A}[a_i] = r[a_i] - \text{mean}(\{r[a_j]\}_{j=1}^{G})$
    Initialize relative policy gradient $\mathbf{g} \leftarrow \mathbf{0}$
    **for** each sampled action $a_i$ where $i = 1$ to $G$ **do**
        $\mathbf{g}[a_i] \leftarrow \mathbf{g}[a_i] + (1 - \pi[a_i]) \cdot \hat{A}[a_i]$
        **for** each other action $a_j \neq a_i$ **do**
            $\mathbf{g}[a_j] \leftarrow \mathbf{g}[a_j] - \pi[a_j] \cdot \hat{A}[a_i]$
        **end for**
    **end for**
    Apply Adam update: $\mathbf{z} \leftarrow \mathbf{z} + \eta \cdot \mathbf{g}/G$
**end for**
**return** Optimized policy parameters $\mathbf{z}$

---

### C.2   HYPERPARAMETER SETTINGS

Our experimental configuration follows that of Zhu et al. [91].

**Training setup**. The prompt batch size is set to 1,024, with 8 rollouts generated per prompt. During training, the sampling temperature is set to 1.0. The maximum context length is configured as 4,096 tokens for both `Qwen2.5-Math-7B` and `Llama-3.2-3B-Instruct`. Model updates are performed with a mini-batch size of 256 and a learning rate of $1 \times 10^{-6}$. For all algorithms, a KL penalty term is incorporated into the final loss function, using a coefficient of $1 \times 10^{-3}$. The clip ratio is set to 0.2. Additionally, an entropy bonus is applied to all objectives with a coefficient of $1 \times 10^{-4}$. All experiments are conducted on a single node with 4 NVIDIA A100 GPUs.

**Evaluation setup**.   During evaluation, we sample 256 responses per prompt for both `Qwen2.5-Math-7B` and `Llama-3.2-3B-Instruct` using a temperature of 0.6 and a top-$p$ value of 0.95. Since the test sets of ARC-c (1,170) and MMLU-Pro (12,000) are relatively large, and sampling 256 times requires substantial computation time, we randomly selected 128 questions and repeated the test three times to obtain the average.

**Prompt template**. Our primary objective is to validate theoretical findings; therefore, a uniform prompt [82] was sampled for all models:

```
<|im_start|>system
You are a helpful assistant.<|im_end|>
<|im_start|>user
{input}
Please reason step by step, and put your final answer within \boxed{}.
<|im_end|>
<|im_start|>assistant
```

## C.3 MORE EVALUATION RESULTS

**Training dynamics of LLama-3.2-3B-Instruct**. The larger model is inherently more capable of retaining broad knowledge and skills during specialized training.

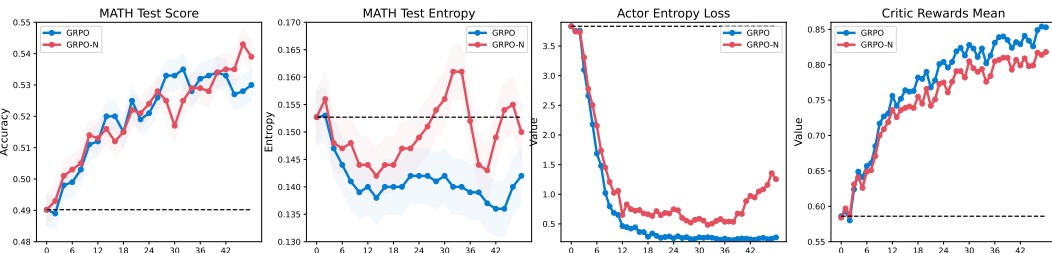

Figure 4: Comparison of the training dynamics of GRPO, GRPO-N on the MATH benchmark across training steps, using the `LLama-3.2-3B-Instruct` model with a prompt batch size of 1,024. **Left Part:** (Left) the greedy decoding accuracy on the MATH test set and (Center Left) the model's entropy on the MATH test set. **Right Part:** (Center Right) the actor entropy loss and (Right) critic rewards mean during training.

Table 2: Pass@$k$ of `Llama-3.2-3B-Instruct` on AMC 2023, AIME 2024, AIME 2025. For each $k$, **bold** and underlined numbers indicate the best and second-best results, respectively.

| Algorithm | | | | Pass@$k$ | | | | | |
|---|---|---|---|---|---|---|---|---|---|
| $k$ | 1 | 2 | 4 | 8 | 16 | 32 | 64 | 128 | 256 |
| **AMC 2023** | | | | | | | | | |
| Base Model | 23.4 | 34.3 | 47.7 | **61.7** | **74.4** | **84.7** | **92.1** | **96.8** | **100.0** |
| GRPO | **31.1** | **41.7** | 51.3 | 58.7 | 64.7 | 70.7 | 76.9 | 83.0 | 87.5 |
| GRPO-N | 30.3 | 41.6 | **52.4** | 60.8 | 67.5 | 74.2 | 81.0 | 87.4 | 92.5 |
| **AIME 2024** | | | | | | | | | |
| Base Model | 6.9 | 11.5 | 17.5 | 23.8 | 29.4 | 33.7 | **37.5** | **42.7** | **50.0** |
| GRPO | 15.7 | 20.6 | 25.1 | 29.1 | 32.2 | 34.4 | 36.1 | 37.9 | 40.0 |
| GRPO-N | **16.2** | **21.2** | **25.8** | **29.9** | **33.2** | **35.2** | 37.3 | 40.8 | 46.7 |
| **AIME 2025** | | | | | | | | | |
| Base Model | 0.4 | 0.9 | 1.7 | 3.2 | 5.6 | 9.2 | 14.6 | **23.2** | **36.7** |
| GRPO | **0.6** | **1.1** | **2.1** | 3.8 | 6.2 | 9.0 | 11.7 | 14.4 | 16.7 |
| GRPO-N | 0.5 | 1.0 | 2.0 | **3.8** | **6.6** | **10.7** | **15.5** | 20.6 | 26.7 |

Table 3: Evaluation results of `Qwen2.5-Math-7B` on MATH-500. For each $k$, **bold** and underlined numbers indicate the best and second-best results, respectively.

| Algorithm | | | | Pass@$k$ | | | | | |
|---|---|---|---|---|---|---|---|---|---|
| $k$ | 1 | 2 | 4 | 8 | 16 | 32 | 64 | 128 | 256 |
| **MATH-500** | | | | | | | | | |
| Base Model | 40.7 | 51.5 | 58.9 | 64.1 | **68.5** | **72.9** | **77.9** | **83.2** | **88.0** |
| GRPO | 53.3 | 57.9 | 61.2 | 63.5 | 65.3 | 67.0 | 68.9 | 70.8 | 72.6 |
| GRPO-N | 53.0 | 57.9 | 61.3 | 63.7 | 65.3 | 66.8 | 68.5 | 70.2 | 72.2 |
| GSPO | 53.0 | 57.7 | 61.0 | 63.4 | 65.2 | 66.8 | 68.6 | 70.5 | 72.8 |
| GSPO-N | **54.1** | **58.8** | **62.0** | **64.1** | 65.9 | 67.6 | 69.5 | 71.5 | 73.4 |

## C.4 DISCUSSION ON RL TRICKS

We also review some widely adopted RL tricks, such as: increasing the number of rollout samples, raising the training temperature, more than three actions case.

● **The number of rollout samples**: a larger $G$ leads to more stable optimization and does not affect our main findings and conclusions, the two-stage dynamic of exploitation and exploration.

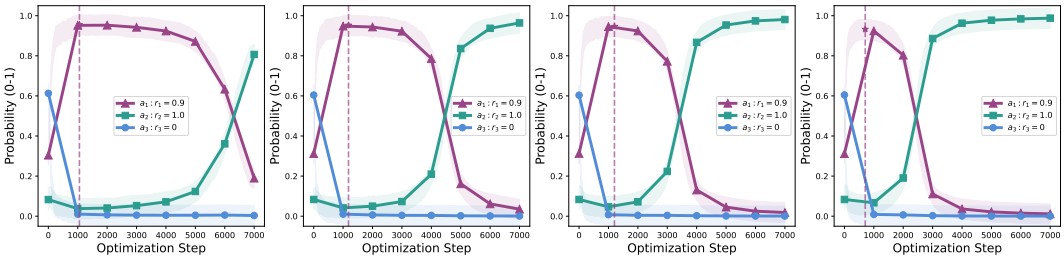

Figure 5: Dynamics of the policy probability mass during optimization for different numbers of rollout samples ($[2, 3, 5, 10]$), with action rewards $r$ and initial policy probabilities $\pi$ held constant.

● **Raising the training temperature**: according to An et al. [2], increasing the sampling temperature enhances the diversity of generated outcomes. Consequently, employing a higher temperature is advisable to obtain a more varied set of trajectories for model training. The default temperature value in our other experiments is $\tau = 1.0$, that is $\pi(\cdot) = \text{Softmax}(\mathbf{z}/\tau)$.

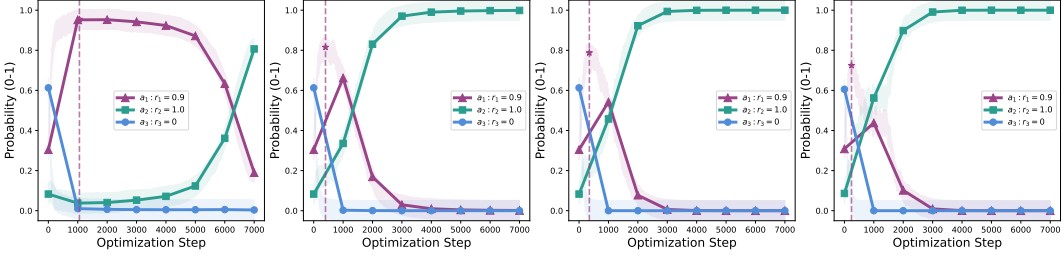

Figure 6: Dynamics of the policy probability mass during optimization for different training temperature values ($[1, 2, 5, 20]$), with action rewards $r$ and initial policy probabilities $\pi$ held constant.

● **More than three actions case**: from Theorem 1, we have: $\mathbb{E}(\Delta z(a_i)) = \eta\pi(a_i)\left[(1 - \pi(a_i))\hat{A}(a_i) - \sum_{j\neq i}^{4} \pi(a_j)\hat{A}(a_j)\right]$.

Denote the action with the largest $r$ as $a_{\max}$ and the action with the smallest $r$ as $a_{\min}$. It can be readily shown that $\mathbb{E}(\Delta z(a_{\max}))$ is always greater than or equal to 0, while $\mathbb{E}(\Delta z(a_{\min}))$ is consistently less than 0. For other actions, the probabilities generally exhibit a two-stage dynamic.

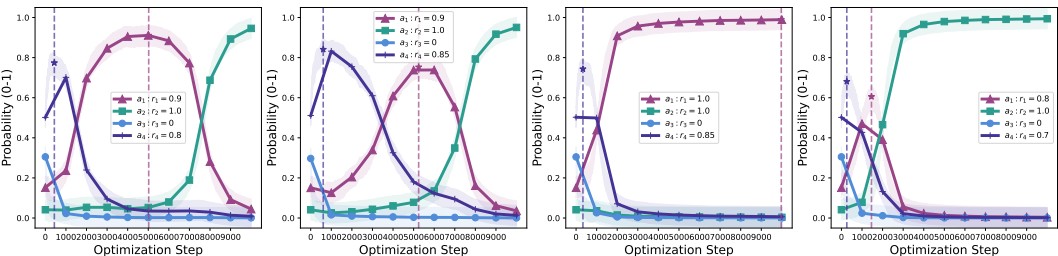

Figure 7: Dynamics of the policy probability mass during optimization for action space consists of four actions, with action rewards $r$ and initial policy probabilities $\pi$ held constant.

## C.5 A FINER-GRAINED METHOD

While current methods for credit assignment in reasoning LLMs (e.g., GRPO, DAPO) are simple and stable, a key limitation is their use of sequence-level feedback. This provides equal credit to all tokens, which, as Section 4.3 demonstrates, weakens token-level guidance and may reinforce the model to treat some errors as acceptable. To achieve more stable and efficient credit assignment while avoiding issues of task independence and the need for a step reward model, we leverage the emergence of stable reasoning prefixes [35, 61] (implicit prefix tree[7]):

**The tree structure arises directly from a set of sampled responses and should not be confused with tree-based sampling, where the tree is predefined.**

As defined in Section 3, each path through the tree traces a complete response from the policy, and every node constitutes a token prefix $y_t$ up to timestep $t$. A branch forms at any token where subsequent responses differ. Terminal nodes receive a binary reward $(0/1)$ depending on verifiable correctness. Following Tran et al. [61], we estimate a token-level value $V(y_t)$ for a given prefix $y_t$:

$$V(y_t) = \frac{1}{|S(y_t)|} \sum_{k \in S(y_t)} r_i,$$

this value is estimated by averaging the normalized rewards across all descendant completions that share this prefix. Here, $S(y_t)$ represents the set of all responses passing through prefix node $y_t$, and $r_i$ denotes the outcome reward of the $i$-th response. Therefore, we can get token-level temporal-difference corrections ($\delta_{i,t} = V(y_{t+1}) - V(y_t)$ for trajectory $i$) derived from the tree. It is crucial to note that $\delta_{i,t}$ deviates from zero exclusively at branching points, as non-branching tokens share identical descendant outcomes, resulting in $V(y_{t+1}) = V(y_t)$. Thus, the final advantage is:

$$\hat{A}_{i,t} = \frac{1}{\text{std}(r)}[r_i - \text{mean}(r) + \delta_{i,t}], \tag{5}$$

where $\frac{1}{\text{std}(r)}[r_i - \text{mean}(r)]$ is the response-level signal GRPO and its variants used, and the $\delta_{i,t}$ provides the token-level guidance for finer-grained credit assignment.

**Last but not the least**, to prolong training and avoid over-exploitation (Remark 3 & Theorem 1), we decay the gradient signals of **sub-paths with no branching nodes and positive advantages** by a factor of 0.1. We denote the method as GRPO-TN. For example,

```
Given the token sequences of three response samples
-Sample 0: [A, B, C, D],
-Sample 1: [A, B, E, F],
-Sample 2: [A, G, H, I]
-with corresponding total scores of 1, 0, and 1 respectively.
We compute token-level advantages according to Equation 5.
The resulting advantage values are:
-Sample 0: [0.707, 0.353, 1.768, 1.768],
-Sample 1: [0.707, -1.768, -2.475, -2.475],
-Sample 2: [0.707, 1.414, 1.414, 1.414].
Notably, we apply attenuation factor to the
subsequences C, D in Sample 0  and G, H, I in Sample 2.
```

---

[7]Li et al. [35] find that despite the variation in final solutions, the generated trajectories share extensive overlapping segments, particularly in the early and intermediate stages of reasoning.

Table 4: Pass@$k$ of `Qwen2.5-Math-7B` on AMC 2023, AIME 2024, AIME 2025. For each $k$, **bold** and underlined numbers indicate the best and second-best results, respectively.

| Algorithm | | | | | Pass@$k$ | | | | |
|---|---|---|---|---|---|---|---|---|---|
| $k$ | 1 | 2 | 4 | 8 | 16 | 32 | 64 | 128 | 256 |
| **AMC 2023** | | | | | | | | | |
| Base Model | 40.4 | 55.6 | 69.1 | 79.4 | 85.9 | 89.5 | 92.1 | 94.6 | 97.5 |
| GRPO | 60.4 | 69.9 | 77.4 | 82.9 | 86.7 | 89.4 | 91.7 | 94.7 | 97.5 |
| GRPO-N | 59.2 | 68.7 | 76.3 | 82.7 | 87.6 | **92.3** | **96.3** | **99.1** | **100.0** |
| GSPO | 61.1 | 70.5 | 78.0 | 83.9 | 88.1 | 91.6 | 94.4 | 96.2 | 97.5 |
| GSPO-N | 61.5 | **71.2** | **78.5** | **84.1** | 88.4 | 91.8 | 94.8 | 97.4 | 100.0 |
| GRPO-TN | **61.6** | 70.7 | 77.7 | 84.0 | **88.7** | 92.0 | 95.8 | 98.4 | 100.0 |
| **AIME 2024** | | | | | | | | | |
| Base Model | 13.6 | 21.8 | 30.5 | 37.5 | 43.5 | 49.7 | 55.8 | 61.4 | 66.7 |
| GRPO | 22.6 | 31.5 | 39.5 | 46.2 | 51.9 | 57.3 | 62.9 | 68.9 | 73.3 |
| GRPO-N | 23.6 | 33.4 | 41.8 | 47.5 | 51.9 | 56.7 | 61.8 | 67.3 | 73.3 |
| GSPO | **25.3** | **34.7** | **42.4** | 48.3 | 53.6 | 58.7 | 63.6 | 68.1 | 73.3 |
| GSPO-N | 23.3 | 31.1 | 42.1 | **48.8** | 54.3 | **59.4** | 64.4 | 69.6 | 73.3 |
| GRPO-TN | 23.8 | 32.8 | 43.2 | 48.2 | **54.9** | 58.7 | **65.1** | **70.2** | **80.0** |
| **AIME 2025** | | | | | | | | | |
| Base Model | 6.4 | 10.2 | 14.5 | 18.9 | 23.6 | 28.1 | 32.5 | 38.3 | 46.7 |
| GRPO | 9.2 | 13.4 | 17.9 | 22.4 | 26.4 | 29.9 | 33.9 | 39.1 | 46.7 |
| GRPO-N | 9.5 | 14.2 | 19.1 | 23.8 | 28.7 | 34.2 | **41.6** | **52.5** | **66.7** |
| GSPO | 9.6 | 14.3 | 19.2 | 23.9 | 28.8 | 34.2 | 40.9 | 49.5 | 60.0 |
| GSPO-N | **10.2** | **14.7** | **19.6** | **24.9** | **29.9** | **35.0** | 40.9 | 47.3 | 53.3 |
| GRPO-TN | 9.6 | 14.3 | 19.3 | 24.3 | 28.9 | 34.3 | 41.1 | 48.3 | 60.0 |

## C.6 ENTROPY BEHAVIOR ANALYSIS FROM DIFFERENT LEVELS

**Set up.** To investigate how different algorithms reshapes the sampling distribution, we compare the base model with the RLVR trained model (using the experimental setup detailed in Section 4).

Following Wu et al. [68], we quantify changes in the output distribution using two entropy metrics:

• Answer-Level Entropy: Let $\{o^{(1)}, \ldots, o^{(G)}\}$ represent the answers extracted from each generated sequence $\mathbf{y}_i$ (with NA denoting incomplete or invalid outputs), and let $\{o_1^*, \ldots, o_M^*\}$ be the set of $M$ distinct answers. Denote by $f_j$ the frequency of answer $o_j^*$, and define the empirical probability as $p_j = \frac{f_j}{G}$. The answer-level entropy is then defined as: $\text{AnswerEntropy} = -\sum_{j=1}^{M} p_j \log p_j$. This metric quantifies the global diversity across output completions, where lower entropy values indicate a greater degree of answer-level certainty.

• Token-Level Entropy: Let $\mathcal{V}$ denote the vocabulary and $\mathbf{y}_i = (y_{i,1}, y_{i,2}, \ldots, y_{i,T})$ denote the $i$-th generated sequence of length $T$ for $1 \leq i \leq N$. At each timestep $t$, the model outputs a probability distribution $p_t^{(i)}(v)$ over vocabulary tokens $v \in \mathcal{V}$. The entropy of this distribution is given by: $H(p_t^{(i)}) = -\sum_{v \in \mathcal{V}} p_t^{(i)}(v) \log p_t^{(i)}(v)$. The average token-level entropy over all $G$ sequences and all timesteps is then computed as: $\text{TokenEntropy} = \frac{1}{G} \frac{1}{T} \sum_{i=1}^{G} \sum_{t=1}^{T} H(p_t^{(i)})$.

Table 5: Summary of entropy metrics across math reasoning benchmarks.

| Metric | Model | AMC 2023 | AIME 2024 | AIME 2025 |
|---|---|---|---|---|
| Answer-Level Entropy | `Qwen2.5-Math-7B` | 2.563 | 4.263 | 4.904 |
| | GRPO | 1.667 | 3.691 | 4.916 |
| | GRPO-N | 1.741 | 3.748 | 4.957 |
| | GSPO | 1.641 | 3.484 | 4.878 |
| | GSPO-N | 1.696 | 3.712 | 4.943 |
| | `Llama-3.2-3B-Instruct` | 3.937 | 5.333 | 6.062 |
| | GRPO | 2.513 | 2.888 | 3.694 |
| | GRPO-N | 2.633 | 3.042 | 4.370 |
| Token-Level Entropy | `Qwen2.5-Math-7B` | 0.118 | - | 0.134 |
| | GRPO | 0.063 | - | 0.054 |
| | GRPO-N | 0.124 | - | 0.092 |
| | GSPO | 0.087 | - | 0.065 |
| | GSPO-N | 0.095 | - | 0.075 |
| | `Llama-3.2-3B-Instruct` | 0.127 | - | 0.107 |
| | GRPO | 0.123 | - | 0.089 |
| | GRPO-N | 0.105 | - | 0.107 |

## C.7 FULL PROCEDURE FOR CASE STUDY

A comparison of the correct responses of GRPO and GRPO-N in mathematical reasoning task (a test case from AMC 2023). The base model is `Qwen2.5-Math-7B`, which is trained on the training set of MATH dataset.

**GRPO:** It exhibits a tendency to repeatedly make similar errors, such as consistently generating code with TypeError issues, and struggles to correct them.

```
To solve the problem, we need to find the maximum value of the
imaginary part of the complex number \(z\) that satisfies the
equation \(|1 + z + z^2| = 4\). Let's break down the problem step-by-step.

1. Express \(z\) in terms of its real and imaginary parts:
   Let \(z = x + yi\), where \(x\) and \(y\) are real numbers,
   and \(i\) is the imaginary unit.
   Then \(z^2 = (x + yi)^2 = x^2 - y^2 + 2xyi\).

2. Substitute \(z\) and \(z^2\) into the given equation:
```

$$
|1 + z + z^2| = |1 + (x + yi) + (x^2 - y^2 + 2xyi)|
$$
$$
= |(1 + x + x^2 - y^2) + (y + 2xy)i| = 4
$$

The magnitude of a complex number $(a + bi)$ is given by $(\sqrt{a^2 + b^2})$. So we have:
$$
\sqrt{(1 + x + x^2 - y^2)^2 + (y + 2xy)^2} = 4
$$
Squaring both sides, we get:
$$
(1 + x + x^2 - y^2)^2 + (y + 2xy)^2 = 16
$$

3. Maximize the imaginary part $(y)$:
   We need to find the maximum value of $(y)$
   that satisfies the above equation.
   Let's use Python and SymPy to solve this.

```python
import sympy as sp

# Define variables
x, y = sp.symbols('x y', real=True)

# Define the equation
equation = (1 + x + x**2 - y**2)**2 + (y + 2*x*y)**2 - 16

# Solve the equation for y in terms of x
solutions = sp.solve(equation, y)

# Find the maximum value of y
max_y = max([sp.simplify(sol) for sol in solutions])

# Simplify the maximum value of y
max_y_simplified = sp.simplify(max_y)

# Extract m and n from the simplified form of max_y
m, n = sp.fraction(max_y_simplified**2)

# Since m and n are relatively prime, we can directly sum them
result = m + n

print(result)
```
```output
TypeError: cannot determine truth value of Relational
```
The error occurs because SymPy's `solve` function
can return symbolic solutions that include inequalities
or other relational expressions.
To handle this, we need to ensure that we are only
considering real solutions and then find the maximum value of $(y)$.

Let's refine the approach by solving the equation numerically
and then finding the maximum value of $(y)$.
We can use SymPy's `nsolve` function to find
numerical solutions and then determine the maximum value of $(y)$.

Here's the revised code:

```python
import sympy as sp

# Define variables
x, y = sp.symbols('x y', real=True)

# Define the equation
equation = (1 + x + x**2 - y**2)**2 + (y + 2*x*y)**2 - 16

# Find critical points by solving the equation
critical_points = sp.solve(equation, y)

# We need to find the maximum value of y
max_y = None
for point in critical_points:
    # Check if the solution is real
    if point.is_real:
        if max_y is None or point > max_y:
            max_y = point

# Simplify the maximum value of y
max_y_simplified = sp.simplify(max_y)

# Extract m and n from the simplified form of max_y
m, n = sp.fraction(max_y_simplified**2)

# Since m and n are relatively prime, we can directly sum them
result = m + n

print(result)
```
```output
TypeError: unsupported operand type(s) for ** or pow(): 'NoneType' and 'int'
```
The error indicates that the `max_y` variable is `None`,
which means that the solutions returned
by `sp.solve` did not yield any real values for $y$.
This suggests that we need to handle the solutions
more carefully and ensure that we are only considering real solutions.

Let's use a different approach to find the maximum value of $y$.
We can use numerical methods to find the maximum value of $y$
by evaluating the solutions over a range of $x$ values.

Here's the revised code using numerical methods:

```python
import sympy as sp
import numpy as np

# Define variables
x, y = sp.symbols('x y', real=True)

# Define the equation
equation = (1 + x + x**2 - y**2)**2 + (y + 2*x*y)**2 - 16

# Function to find the maximum value of y for a given x
```

```
def find_max_y():
    max_y = None
    for x_val in np.linspace(-10, 10, 1000):
        solutions = sp.solve(equation.subs(x, x_val), y)
        for sol in solutions:
            if sol.is_real:
                if max_y is None or sol > max_y:
                    max_y = sol
    return max_y

# Find the maximum value of y
max_y = find_max_y()

# Simplify the maximum value of y
max_y_simplified = sp.simplify(max_y)

# Extract m and n from the simplified form of max_y
m, n = sp.fraction(max_y_simplified**2)

# Since m and n are relatively prime, we can directly sum them
result = m + n

print(result)
```
``` output
timeout_decorator.timeout_decorator.TimeoutError: 'Timed Out'
```
The numerical approach can be computationally intensive and
may not be the most efficient way to solve this problem.
Instead, let's use a more analytical approach to find the maximum value of $y$.

We can use the fact that the equation $|1 + z + z^2| = 4$ represents
a certain geometric shape in the complex plane.
To find the maximum value of the imaginary part of $z$,
we can use the properties of complex numbers and their magnitudes.

Let's rewrite the equation in a more manageable form. We have:
$$
|1 + z + z^2| = 4
$$
This can be written as:
$$
|1 + x + yi + (x + yi)^2| = 4
$$
Simplifying the expression inside the magnitude, we get:
$$
|1 + x + yi + (x^2 - y^2 + 2xyi)| = 4
$$
$$
|1 + x + x^2 - y^2 + (y + 2xy)i| = 4
$$
The magnitude of a complex number $(a + bi)$
is given by $(\sqrt{a^2 + b^2})$. So we have:
$$
\sqrt{(1 + x + x^2 - y^2)^2 + (y + 2xy)^2} = 4
$$
Squaring both sides, we get:
$$
(1 + x + x^2 - y^2)^2 + (y + 2xy)^2 = 16
$$

\]
To find the maximum value of \(y\), we can use calculus to find
the critical points of this equation.
However, a more straightforward approach is to recognize
that the maximum value of \(y\) will occur
when the imaginary part of \(z\) is maximized.

By analyzing the equation, we can see that the maximum value of \(y\)
occurs when the real part of the expression inside the magnitude
is minimized and the imaginary part is maximized.
This happens when \(x = -\frac{1}{2}\), which simplifies the equation to:
\[
\left(1 - \frac{1}{2} + \left(-\frac{1}{2}\right)^2
- y^2\right)^2 + (y + 2\left(-\frac{1}{2}\right)y)^2 = 16
\]
Simplifying further, we get:
\[
\left(1 - \frac{1}{2} + \frac{1}{4} - y^2\right)^2 + (y - y)^2 = 16
\]
\[
\left(\frac{3}{4} - y^2\right)^2 = 16
\]
Taking the square root of both sides, we get:
\[
\left|\frac{3}{4} - y^2\right| = 4
\]
This gives us two cases:
1. \(\frac{3}{4} - y^2 = 4\)
2. \(\frac{3}{4} - y^2 = -4\)

Solving the first case:
\[
\frac{3}{4} - y^2 = 4
\]
\[
y^2 = \frac{3}{4} - 4 = -\frac{13}{4}
\]
This is not a valid solution since \(y^2\) cannot be negative.

Solving the second case:
\[
\frac{3}{4} - y^2 = -4
\]
\[
y^2 = \frac{3}{4} + 4 = \frac{19}{4}
\]
So the maximum value of \(y\) is:
\[
y = \sqrt{\frac{19}{4}} = \frac{\sqrt{19}}{2}
\]
Thus, the maximum value of the imaginary part of \(z\)
is \(\frac{\sqrt{19}}{2}\).
The values \(m\) and \(n\) are 19 and 2, respectively,
and they are relatively prime. Therefore, \(m + n = 19 + 2 = 21\).

The final answer is:
\[
\boxed{21}
\]

**GRPO-N:** It exhibits a reduction in erroneous code generation relative to GRPO.

To solve the problem, we need to find the maximum imaginary part of a complex number $z$ that satisfies $|1 + z + z^2| = 4$. Let's break down the problem into manageable steps:

1. Express $z$ in terms of its real and imaginary parts:
Let $z = x + yi$, where $x$ and $y$ are real numbers.
2. Substitute $z$ into the given equation:
The equation becomes $|1 + (x + yi) + (x + yi)^2| = 4$.
Simplify the expression inside the absolute value:
$$
1 + (x + yi) + (x + yi)^2 = 1 + x + yi + x^2 - y^2 + 2xyi
= (1 + x + x^2 - y^2) + (y + 2xy)i
$$
The magnitude of this complex number is:
$$
\sqrt{(1 + x + x^2 - y^2)^2 + (y + 2xy)^2} = 4
$$
3. Square both sides to remove the square root:
This gives us the equation:
$$
(1 + x + x^2 - y^2)^2 + (y + 2xy)^2 = 16
$$
4. Find the maximum value of $y$:
We need to maximize $y$ subject to the above equation.
We can use calculus or numerical methods to find the maximum value of $y$,
but for now, let's proceed with a numerical approach
using Python to find the maximum value of $y$.

We'll use a grid search method to find the maximum value of $y$.
We'll iterate over a range of $x$ and $y$ values and
heck which values satisfy the equation.
We'll keep track of the maximum $y$ value found.

Let's write the Python code to perform this search.
```python
import numpy as np

# Define the function that represents the equation
def equation(x, y):
    return (1 + x + x**2 - y**2)**2 + (y + 2*x*y)**2 - 16

# Initialize variables to keep track of the maximum y value
max_y = -np.inf
max_x = 0

# Define the range and step size for x and y
x_vals = np.linspace(-5, 5, 500)
y_vals = np.linspace(-5, 5, 500)

# Iterate over the range of x and y values
for x in x_vals:
    for y in y_vals:
        if np.isclose(equation(x, y), 0, atol=1e-5):
            if y > max_y:
                max_y = y
                max_x = x
```

```
1674    print((max_y, max_x))
1675    ```
1676    ```output
1677    (-0.030060120240481286, -1.997995991983968)
1678    ```
1679    The numerical method suggests that the maximum value of $\(y\)$
1680    is approximately $\(-0.03\)$, which is very close to 0.
1681    This indicates that the maximum imaginary part of $\(z\)$ is likely 0,
1682    but let's verify this by checking the exact form of the solution.
1683
1684    To find the exact maximum imaginary part, we can use
1685    the fact that the equation $\(|1 + z + z^2| = 4\)$ represents
1686    a curve in the complex plane.
1687    We can analyze this curve to find the maximum imaginary part analytically.
1688
1689    Let's consider the equation again:
1690    \[
1691    (1 + x + x^2 - y^2)^2 + (y + 2xy)^2 = 16
1692    \]
1693    To maximize $\(y\)$, we can look at the critical points of this equation.
1694    We can use the method of Lagrange multipliers
1695    or directly analyze the equation to find the maximum value of $\(y\)$.
1696
1697    By analyzing the equation, we can see that
1698    the maximum imaginary part occurs when $\(x = -1/2\)$.
1699    Substituting $\(x = -1/2\)$ into the equation, we get:
1700    \[
1701    (1 - \frac{1}{2} + \frac{1}{4} - y^2)^2 + (y - y)^2
1702    = 16 \implies \left(\frac{3}{4} - y^2\right)^2 = 16
1703    \]
1704    Solving for $\(y\)$, we get:
1705    \[
1706    \frac{3}{4} - y^2 = \pm 4 \implies y^2 = \frac{3}{4} \pm 4
1707    \]
1708    Since $\(y^2\)$ must be non-negative, we take the positive root:
1709    \[
1710    y^2 = \frac{3}{4} + 4 = \frac{19}{4} \implies y = \pm \frac{\sqrt{19}}{2}
1711    \]
1712    The maximum imaginary part is $\(\frac{\sqrt{19}}{2}\)$.
1713    Therefore, $\(m = 19\)$ and $\(n = 2\)$, so $\(m + n = 21\)$.
1714
1715    Thus, the final answer is:
1716    \[
1717    \boxed{21}
1718    \]
```

