# OpenReview forum: "The Debate on RLVR Reasoning Capability Boundary: Shrinkage, Expansion, or Both? A Two-Stage Dynamic View"
_ICLR.cc/2026/Conference — Submitted to ICLR 2026_

### Official Review · Reviewer_PFHy · 2025-10-18

**Soundness:** 3
**Presentation:** 3
**Contribution:** 2
**Rating:** 4
**Confidence:** 4

**Summary:**

This paper investigates a central controversy in RLVR: whether RLVR shrinks or expands the reasoning capability boundary of LLMs. By analyzing the dynamical evolution of the change of logits in the gradient space, the authors propose a two-stage dynamic framework of probability mass evolution:

1. Exploitation stage: the model mainly reinforces high-reward tokens, causing capability boundary shrinkage.
2. Exploration stage: once high-probability tokens saturate, relative negative gradients redirect updates to low-probability yet potentially optimal tokens, enabling boundary expansion.

They derive the expected logit update rule (Theorem 1), validate the two-stage behavior through a toy example, and then empirically evaluate GRPO-N and GSPO-N (variants using only relative negative gradients) on MATH, AIME, AMC, ARC-c, and MMLU-Pro benchmarks, showing improved exploration and stable Pass@k performance compared to GRPO/GSPO.

The topic is timely and relevant, the proposed two-stage framework is inspiring and well supported by toy example and theoretical analysis. However, I have the following two main concerns. First is the potential gap between theoretical analysis and (see weakness 1) the practical GRPO methods we use. Second is the novelty of the proposed method (see weakness 2). So, although I indeed enjoy reading this paper and believe that the paper has great potential, I could only give a weak reject at this stage. I am looking forward to see the improvement of the paper and would be happy to increase my evalution during the discussion stage.

**Strengths:**

1. **Timely and relevant topic.**
The “RLVR reasoning capability debate” is currently a major research question (e.g., DeepSeek-R1, O1/O3). The paper addresses it with both theoretical and empirical clarity.

2. **Elegant theoretical framing and well-designed toy example**
The derivation of *Lemma 1 + Theorem 1* connects token-level probability dynamics to capability boundary movement, presenting a physically interpretable “force-like” view similar to learning-dynamics literature. The intuition that action a1 saturates and then the confidence of action a2 gradually increases is very novel and inspiring. (However, this also leads to my main concern in weakness 1).

3. **The paper is well written and easy to follow.**

**Weaknesses:**

1. If I understand correctly, the two-stage dynamics unfold as follows, following the notations from the toy example. At the beginning of training, action a1 (with reward r1=0.8) and action a3 (with reward r3=0) have relatively high confidence and are therefore more likely to be sampled than action a2 (with reward r2=1.0). During this early stage, since the confidence on a1 has not yet saturated, even though the model occasionally samples a2 by chance, the increase in the probability of a1 dominates the updates. In other words, the probability mass reduced from penalizing a3 is primarily reallocated to a1.

    As training proceeds and the probability of a1 gradually saturates while the probability of a3 continues to decrease, the probability mass released from a3 is then absorbed by a2. This marks the beginning of the exploration stage described in the paper, where the pair of a1 and a2 may be sampled within the same group. In this phase, since r1<r2, a1 effectively becomes the relative negative example, and the probability mass shifts from a1 to a2. This process can, in principle, continue if there exist additional hidden actions, for example, a4 with reward r4>r2>r1. (I.e., a4 will dominate as the training goes on.)

    **A key assumption in the above reasoning is that r1<r2.** If r1=r2, the gradients applied to a1 and a2 would always be identical, preventing a2 from absorbing probability mass from a1. However, in most practical implementations of GRPO and its variants, the reward is 1/0. This creates a conceptual gap between the theoretical framework and real training settings.
    That said, I do not view this gap as a bad thing. Rather, it highlights an important research direction: incorporating more fine-grained reward structures into RLVR training. From this perspective, I find the framework proposed in this paper highly promising. If the authors could explicitly discuss this assumption and consider RLVR variants with continuous or graded rewards, the paper would be significantly strengthened.

2. The proposed method itself is not sufficiently novel. GRPO-N and GSPO-N simply remove the positive-gradient component, which is conceptually similar to the NSR configuration discussed in [1]. I understand that the main contribution of this paper lies in introducing the two-stage framework to interpret the dynamics of reasoning capability in RLVR, rather than proposing a new algorithm. However, incorporating an algorithmic idea directly inspired by this framework would make the paper much stronger. For instance, one could design a quantitative metric to detect the end of the exploitation stage during training, and subsequently increase the influence of negative examples once the model transitions into the exploration stage.

[1] Zhu, Xinyu, et al. "The surprising effectiveness of negative reinforcement in LLM reasoning." arXiv preprint arXiv:2506.01347 (2025).

**Questions:**

1. It would be nice to add some titles to each subfigure in Figure 1. Or at least highlight their different settings in the caption. It took me quite a while to find that the rewards of different settings (in small legends) are different.

2. The second panel in Figure 2 shows that GSPO can mitigate the entropy decay compared with GRPO. The gap after ablating the positive gradients of GSPO is also smaller. Will the theory proposed in this paper explain this phenomenon?

3. I am not quite sure what we could conclude from Figure 3. Any discussions about that?

4. How sensitive is the two-stage transition point to group size G?

5. (Not so important) Is it possible to have a demonstration in the real setting, like Figure 1 in the toy example? Figure 2 might be a good starting point. Maybe directly observing the confidence of the chosen token and other candidate tokens? Not quite sure.

---

> ### Author Response · Authors · 2025-11-14
> **Thank you! (part 1)**
>
> Dear Reviewer PFHy,
>
> We **deeply appreciate your encouraging recognition** of our theoretical framing and the novel dynamics. Your comments are highly appreciated, and we have addressed each point in the following responses. In response to all comments, we have revised the manuscript and marked the changes in **orange**.
>
> > **W1**: If I understand correctly, the two-stage dynamics unfold as follows, following the notations from the toy example. At the beginning of training, action a1 (with reward r1=0.8) and action a3 (with reward r3=0) have relatively high confidence and are therefore more likely to be sampled than action a2 (with reward r2=1.0). During this early stage, since the confidence on a1 has not yet saturated, even though the model occasionally samples a2 by chance, the increase in the probability of a1 dominates the updates. In other words, the probability mass reduced from penalizing a3 is primarily reallocated to a1.
> >
> > As training proceeds and the probability of a1 gradually saturates while the probability of a3 continues to decrease, the probability mass released from a3 is then absorbed by a2. This marks the beginning of the exploration stage described in the paper, where the pair of a1 and a2 may be sampled within the same group. In this phase, since r1<r2, a1 effectively becomes the relative negative example, and the probability mass shifts from a1 to a2. This process can, in principle, continue if there exist additional hidden actions, for example, a4 with reward r4>r2>r1. (I.e., a4 will dominate as the training goes on.)
> >
> > A key assumption in the above reasoning is that r1<r2. If r1=r2, the gradients applied to a1 and a2 would always be identical, preventing a2 from absorbing probability mass from a1. However, in most practical implementations of GRPO and its variants, the reward is 1/0. This creates a conceptual gap between the theoretical framework and real training settings. That said, I do not view this gap as a bad thing. Rather, it highlights an important research direction: incorporating more fine-grained reward structures into RLVR training. From this perspective, I find the framework proposed in this paper highly promising. If the authors could explicitly discuss this assumption and consider RLVR variants with continuous or graded rewards, the paper would be significantly strengthened.
>
> > **A**: We are truly delighted and honored by your insightful comments. You have perfectly grasped the core two-stage dynamic we aimed to illustrate, and we are especially pleased that **your understanding of points (1) and (2) is entirely accurate**. A pivotal aspect of this process is indeed the shifting identity of the "relative negative example" between training stages, as you correctly identified. Furthermore, we **have also discussed in the Appendix C.4 the scenario involving additional hidden actions** with even higher rewards. Your positive assessment is a great encouragement to us.
> >
> > For point (3), it's true that "If r1=r2, the gradients applied to a1 and a2 would always be identical, preventing a2 from absorbing probability mass from a1." **This may also be an issue present in NSR** [1]. In NSR (their Appendix D.3) "we disable this normalization in PSR and NSR’s implementation" and "the advantage is equal to the raw reward (i.e., +1 for PSR and −1 for NSR)." Therefore, more precisely, the NSR represents **negative-reward gradients** and may **not account for relative changes** during the training process.  See Section 6 in the revised version for details.
> >
> > [1] Zhu, Xinyu, et al. "The surprising effectiveness of negative reinforcement in LLM reasoning." arXiv preprint arXiv:2506.01347 (2025).
> >
> > In contrast, GRPO-N uses **negative-advantage gradients** (A=r-mean(r)/std). As discussed in our theoretical analysis, **a key feature of the two-stage dynamic is the shift in relative negative samples**: from initially low-reward tokens (Figure 1 a3) to previously high-reward tokens (Figure 1 a1). Consequently, here, positive reward samples still serve the function of computing relative advantages by providing learning signals. Therefore, in most practical implementations of GRPO and its variants, **different positive advantage actions will also have varying levels of advantage** (continuous or graded advantages).
> >
> > Last but not the least, we fully endorse your view on the importance of fine-grained reward structures in RLVR training. We will discuss this in **A to W2**.

---

> > ### Author Response · Authors · 2025-11-14
> > **Thank you! (part 2)**
> >
> > > **W2**: The proposed method itself is not sufficiently novel. GRPO-N and GSPO-N simply remove the positive-gradient component, which is conceptually similar to the NSR configuration discussed in [1]. I understand that the main contribution of this paper lies in introducing the two-stage framework to interpret the dynamics of reasoning capability in RLVR, rather than proposing a new algorithm. However, incorporating an algorithmic idea directly inspired by this framework would make the paper much stronger. For instance, one could design a quantitative metric to detect the end of the exploitation stage during training, and subsequently increase the influence of negative examples once the model transitions into the exploration stage.
> >
> > > **A**: We sincerely thank the reviewer for raising this important point.
> > >
> > > (1) We appreciate the opportunity to further summarize and clarify our work's contribution in relation to NSR [1].
> > >
> > > For Differences in Experimental Setup, please see **A to W1** for details.
> > >
> > > For Theoretical contribution: we appreciate that you have rightly recognized the paper's main contribution lies in its two-stage framework for interpreting reasoning dynamics in RLVR, not in introducing a new algorithm. **Guided by these findings, one can develop new algorithms to foster more advanced reasoning capabilities. The GSPO-N strategy serves as a straightforward implementation validating the guiding principles of our theory**.
> > >
> > > (2) Besides, it would be our pleasure to **explore some potential future directions** (Section 6 in the revised version) with you:
> > >
> > > For finer-grained control of probability assignments: as demonstrated in Figure 3 (Section 4.3), notable distinctions exist even among different correct responses: instances where incorrect code was initially produced but later reflected upon and corrected to form the final correct response. GRPO assigns higher probability to the **entire trajectory** (i.e., **the behavior of generating incorrect code is reinforced**—a tendency that may significantly affect the model’s self-correction ability), whereas GRPO-N mitigates this issue.
> > >
> > > Intuitively, we can **filter correct responses** based on the count of correct Python (or tools) call executions they contain. For instance, in E-GRPO [2], a dense entity-aware reward is formulated based on the number of ground-truth entities identified, which by definition creates a **task-dependent reward**.
> > >
> > > To promote diversity, we can employ text similarity metrics [1] to filter the samples. This helps mitigate excessive exploration in the first stage and encourages a transition to the second. **However, the reliance on a step-wise training data construction process** in [1] may lead to increased variance in the training.
> > >
> > > In the future, we will focus on **exploring task-independent approaches** that eliminate the need for explicit step-wise decomposition and scoring.
> > >
> > > (3) Designing a quantitative metric to identify the conclusion of the exploitation phase is a challenging yet crucial endeavor. This difficulty arises from the inherent stochasticity of online sampling in RL algorithms, which is further compounded by the variability across different models and training datasets. However, **we can offer some experimental insights**.  As shown in Figure 2, methods like GRPO-N, GSPO, and GSPO-N all exhibit a characteristic pattern where the entropy initially decreases and then increases. We can leverage the magnitude of this entropy change as a detection metric. Specifically, when the entropy decrease in GRPO slows down, we can subsequently increase the influence of negative examples.
> > >
> > > [1] Supervised Reinforcement Learning: From Expert Trajectories to Step-wise Reasoning. arXiv:2510.25992 (released after our submission).
> > >
> > > [2] Repurposing Synthetic Data for Fine-grained Search Agent Supervision. Tongyi DeepResearch (released after our submission).
> >
> > > **Q1**: It would be nice to add some titles to each subfigure in Figure 1. Or at least highlight their different settings in the caption. It took me quite a while to find that the rewards of different settings (in small legends) are different.
> >
> > > **A**: Thank you for this constructive comment. We will highlight the differences in settings and rewards. This will undoubtedly enhance the readability of the figure.

---

> > ### Author Response · Authors · 2025-11-14
> > **Thank you! (part 3)**
> >
> > > **Q2**: The second panel in Figure 2 shows that GSPO can mitigate the entropy decay compared with GRPO. The gap after ablating the positive gradients of GSPO is also smaller. Will the theory proposed in this paper explain this phenomenon?
> >
> > > **A**: This is a pertinent question. (1) First, we recall the key distinction between GSPO and GRPO, as outlined in Lines 138–142: while the token-level importance ratio w_{i,t}in GRPO may not align well with sequence-level rewards, GSPO introduces a sequence-level importance ratio w_i​ based on sequence likelihood. (2) Moreover, certain tokens (such as “wait”) that frequently appear in both positive and negative rollouts can lead to instability when optimized with token-level importance ratios. As illustrated in Figure 2, this explains why both GSPO and GSPO-N exhibit greater stability compared to GRPO and GRPO-N.
> > >
> > > For theoretical explanation,  from a high-level understanding perspective, you can view the sequence as a holistic action. Therefore, the sequence-level importance ratio
> > ($w_i(\theta)$) mitigates, to some extent, the influence of the initial policy's  $\pi(\cdot\mid \mathbf{x})$ value on the expected policy gradient ( $\mathbb{E}_{\sim \pi(\cdot\mid \mathbf{x})}$).
> > We add this discussion in the revised version (Section 6).
> >
> > > **Q3**: I am not quite sure what we could conclude from Figure 3. Any discussions about that?
> >
> > > **A**: That is a good question! We appreciate the opportunity to clarify this point.
> > >
> > > Under current outcome-only reward schemes (in GRPO),  trajectories with incorrect intermediate python calls can still receive positive reward if the final answer is correct, effectively **reinforcing the model to treat such errors are acceptable**.  This may significantly affect the model’s self-correction ability. In contrast, the model trained with **GRPO-N produces fewer erroneous codes**.
> > >
> > > As we discussed in **A to W2**, this necessitates an inquiry into achieving finer-grained control of probability assignments.
> >
> > > **Q4**: How sensitive is the two-stage transition point to group size G?
> >
> > > **A**: Good question! As discussed in **Appendix C.4**, we have thoroughly investigated the impact of different group sizes G. Our analysis shows that while a larger G contributes to more stable optimization, it does not alter our main findings and conclusions regarding the two-stage dynamic of exploitation and exploration. Overall, we observe that larger G makes it easier to sample potentially optimal correct actions, which aligns with and reinforces our central arguments.
> >
> > > **Q5**: (Not so important) Is it possible to have a demonstration in the real setting, like Figure 1 in the toy example? Figure 2 might be a good starting point. Maybe directly observing the confidence of the chosen token and other candidate tokens? Not quite sure.
> > >
> > > **A**: Thank you for the suggestion. As we analyzed in Lines 58-65, in a real-world setting, the search tree grows exponentially at a rate of $O(V^T)$, where $V$ is the vocabulary size and $T$ is the maximum generation length. Because of this, clearly demonstrating the dynamic changes in a real scenario becomes highly complex.
> > >
> > > However, We also identify several studies conducted in **real-world settings that substantiate our analysis**: [1,2] highlight the significance of high-entropy tokens to RLVR. In contrast, [2] further finds that training exclusively on the low-entropy tokens leads to a marked decline in performance. Intuitively, low-entropy tokens can be viewed as previously explored actions (with high determinism, $\pi$～1), whereas high-entropy tokens represent actions with significant uncertainty (potential actions). Over-optimizing low-entropy tokens may lead to over-exploitation.
> > >
> > > [1] Demystifying Reasoning Dynamics with Mutual Information: Thinking Tokens are Information Peaks in LLM Reasoning. NeurIPS 2025.
> > >
> > > [2] Beyond the 80/20 Rule: High-Entropy Minority Tokens Drive Effective Reinforcement Learning for LLM Reasoning. Qwen Team.

---

> ### Author Response · Authors · 2025-11-21
> **We look forward to your reply (Supplement to A to W1)!**
>
> Dear Reviewer PFHy,
>
> Thank you once more for your review. Please allow us to elaborate **further with additional discussion and exposition**.
>
> > Your main concern in weakness 1: A key assumption in the above reasoning is that r1<r2 ......  From this perspective, I find the **framework proposed in this paper highly promising**. If the authors could explicitly discuss this **assumption and consider RLVR variants with continuous or graded rewards**, the paper would be significantly strengthened.
>
> > A:
> >
> > (1) For the the validity of the assumptions：as we described in Section 2, in most practical implementations of GRPO/GRPO-N, the coefficient $w_{i,t}\hat{A}_{i,t}$ is various among different correct responses. NSR uses only the raw-reward $r_i$. (see **A to W1** for more details)
> >
> > ----
> >
> > (2) For "consider RLVR variants with continuous or graded rewards":
> >
> > **We firmly believe that a good theory should not only account for empirical observations but also provide actionable insights for design**. Under the guidance of our theoretical framework, one can develop novel algorithms to cultivate more sophisticated reasoning capacities.
> >
> > The GRPO-N strategy serves as a straightforward implementation validating the guiding principles of our theory. **Additionally, Appendix C.5 provides a method for finer-grained control of probability assignments, inspired by our theoretical analysis**.
> >
> > In brief (see test scores in the revised version), we leverage the emergence of stable reasoning prefixes—— [1,2] find that despite the variation in final solutions, the generated trajectories share extensive overlapping segments, particularly in the early and intermediate stages of reasoning. The tree structure arises directly from a set of sampled responses and should not be confused with tree-based sampling, where the tree is predefined.
> >
> > As defined in Section 3, each path through the tree traces a complete response from the policy, and every node constitutes a token prefix $y_t$ up to timestep $t$. A branch forms at any token where subsequent responses differ. Terminal nodes receive a binary reward ($0/1$) depending on verifiable correctness. Following \citet{tempo}, we estimate a token-level value $V(y_t)$ for a given prefix $y_t$:  $ V(y_t)=\frac{1}{|S(y_t)|}\sum_{k\in S(y_t)}r_i$,
> this value is estimated by averaging the normalized rewards across all descendant completions that share this prefix. Here, $S(y_t)$ represents the set of all responses passing through prefix node $y_t$, and $r_i$ denotes the outcome reward of the $i$-th response. Therefore, we can get token-level temporal-difference corrections ($\delta_{i,t}=V(y_{t+1})-V(y_t)$ for trajectory $i$) derived from the tree. It is crucial to note that $\delta_{i,t}$ deviates from zero exclusively at branching points, as non-branching tokens share identical descendant outcomes, resulting in $V(y_{t+1})=V(y_t)$.
> >
> > Thus, the final advantage is:
> $\hat{A}_{i,t}$
> >
> > = $\frac{1}{\text{std}(r)}[r_i-\text{mean}(r)+\delta_{i,t}]$,  (*)
> >
> > where $\frac{1}{\text{std}(r)}[r_i-\text{mean}(r)]$ is the response-level signal GRPO and its variants used, and the $\delta_{i,t}$ provides the token-level guidance for finer-grained credit assignment.
> >
> > **Last but not the least, to prolong training and avoid over-exploitation (Remark 3 & Theorem 1), we decay the gradient signals of sub-paths with no branching nodes and positive advantages** by a factor of 0.1. We denote the method as **GRPO-TN**.
> >
> > -----
> > For example, given the token sequences of three response samples
> >
> > -Sample 0: [A, B, C, D],
> >
> > -Sample 1: [A, B, E, F],
> >
> > -Sample 2: [A, G, H, I]
> >
> > -with corresponding total scores of 1, 0, and 1 respectively.
> >
> > We compute token-level advantages according to Equation (*).
> The resulting advantage values are:
> >
> > -Sample 0: [0.707, 0.353, 1.768, 1.768],
> >
> > -Sample 1: [0.707, -1.768, -2.475, -2.475],
> >
> > -Sample 2: [0.707, 1.414, 1.414, 1.414].
> >
> > Notably, we apply attenuation factor to the
> subsequences C, D in Sample 0  and G, H, I in Sample 2.
> >
> > [1] TreePO: Bridging the Gap of Policy Optimization and Efficacy and Inference Efficiency with Heuristic Tree-based Modeling. arXiv:2508.17445.
> >
> > [2] Exploiting tree structure for credit assignment in rl training of llms. arXiv:2509.18314, 2025.
>
> Thank you again for your review. We would be pleased to address any additional questions or concerns you might have. Wishing you every success in your research and personal endeavors!

---

> > ### Comment · Reviewer_PFHy · 2025-11-22
> > **Thanks for your response, still have some concerns about W1.**
> >
> > Thank you for the extensive and thoughtful rebuttal. Many of my earlier concerns have been addressed, and I appreciate the authors’ effort in improving the clarity and organization of the paper.
> >
> > However, the key issue raised in W1 still feels insufficiently resolved. The theoretical framework relies on the assumption r1<r2, while standard RLVR settings typically use binary rewards (1/0), where multiple positive samples may share identical rewards. This creates a conceptual gap between the two-stage dynamics described in the theory and the actual training regime. The additional explanations provided, mainly around token-level advantages and tree-based credit assignment, do not directly clarify whether the two-stage mechanism rigorously holds under the 1/0 reward structure, nor how the theory should be interpreted in such cases.
> >
> > In short, I do not believe that other related works (such as NSR) rely on this assumption as an excuse; the theory and experiments in a paper should be aligned and self-contained. This gap weakens the strength of the theoretical contribution. Given the short rebuttal window, I understand it may be difficult to fully resolve. I encourage the authors to consider addressing this point in a future revision, either by:
> >
> > (1) Extending the theoretical analysis to explicitly accommodate equal-reward cases, or
> > (2) Complementing the theory with experiments under non-binary reward settings to validate the assumptions.
> >
> > That said, the paper has clearly improved in several other aspects, and the authors have made substantial efforts to clarify motivation, structure, and algorithmic implications. Taking these improvements into account, I am inclined to raise my score from 4 to 6.

---

> > > ### Author Response · Authors · 2025-11-22
> > > **Thank you for your kind words and positive feedback (Additional Clarification Regarding W1)！**
> > >
> > > Dear Reviewer PFHy,
> > >
> > > **Please accept our heartfelt thanks for your supportive feedback**. We are delighted that you have noted the improvements in the manuscript's motivation, structure and algorithmic implications. Your constructive engagement throughout this process has been invaluable.
> > >
> > > > Additionally, please allow us to provide some additional clarifications (We will add a detailed description of this in the next version):
> > > >
> > > > ---
> > > >
> > > > (1) We would like to gently clarify a potential misunderstanding. **It was not our intention to claim** that "other related works (such as NSR) rely on this assumption as an excuse." **We aimed to demonstrate** that NSR (a good work, its Appendix D.3, "the advantage is equal to the raw reward 1/0" ) embodies negative-reward gradients and may not capture relative changes in the training process. Consequently, this leads to the phenomenon you accurately described, which "prevents a2 from absorbing probability mass from a1".
> > > >
> > > > ---
> > > >
> > > > (2) For the the validity of the assumptions. Our additional explanations on token-level advantages and tree-based credit assignment were aimed at : **supporting the validity of the assumption and verifying the operational implications of the theory**. Specifically, the gradient in the actual RLVR setup is proportional to $w_{i,t}\hat{A}_{i,t}\nabla \log \pi$, rather than $r_i\nabla \log \pi$. On this basis, the assumption is reasonable, and we are confident that the theoretical and experimental components of our paper are aligned and self-contained.
> > > >
> > > > ---
> > > >
> > > > (3) For whether the two-stage mechanism rigorously holds under the 1/0 reward structure, the answer depends on the specific case: **One the one hand**, If we consider the entire sentence as a complete action sample, and if both actions a1 and a2 receive a reward of 1 (i.e., both are “good” actions), then—as analyzed in Remarks 2 and 3—the update depends on both the advantage estimate A and the current policy distribution π. Under the two-stage dynamic analysis, neither p(a1) nor p(a2) will converge to 0 in the limit. **On the other hand**, if we model each token as an action, a sentence is generated through a sequence of multiple action samplings. Under this perspective, the sampling space expands to $O(V^T)$, as outlined in Section 3. Here, the $w_{i,t}\hat{A}_{i,t}$ for every token is various, the two-stage mechanism still holds, this may be  "under non-binary reward settings to validate the assumptions".
> > > >
> > > > That being said, we are truly thankful for your patience and the constructive criticism, which have been instrumental in helping us improve the quality of our work. In the future, we will **make our best efforts to complement our whole work by rigorously exploring the suggested directions**.
> > > >
> > > We are deeply grateful for your positive feedback and for the time and effort you have invested in your thoughtful review. Your valuable insights have significantly enriched our work. **Wish you all the best!**

---

### Official Review · Reviewer_xPjS · 2025-10-29

**Soundness:** 3
**Presentation:** 3
**Contribution:** 2
**Rating:** 4
**Confidence:** 4

**Summary:**

The paper studies whether RL with verifiable rewards (RLVR) causes reasoning shrinkage or expansion in large language models. It proposes a theoretical analysis of Softmax logit and what to link to the “two-stage dynamics”:
(1) an exploitation stage where high-reward tokens are reinforced and probability mass concentrates, causing capability shrinkage, followed by
(2) an exploration stage where prolonged training redistributes probability mass toward previously low-probability optimal actions, enabling capability expansion.

**Strengths:**

1. Discussing a real and important empirical debate in RLVR reasoning.
2. The toy helps intuition
3. Experiments cover multiple baselines (GRPO, GSPO, ±N variants) and several reasoning benchmarks.

**Weaknesses:**

The central “two-stage dynamics” claim lacks rigorous justification:
- The proposed two-phase interpretation remains largely speculative and is not formally or mathematically derived.
- The analysis does not identify when—or under what precise conditions—a transition between “shrinkage” and “expansion” should occur, nor does it guarantee that such a transition will happen in practice.
- In particular, if the model is already sufficiently exploitative, it may rarely sample low-confidence alternative solutions, making the proposed expansion phase unlikely and potentially leading instead to losing exploration ability.

GRPO-N lacks novelty and is equivalent to prior NSR: The paper’s main algorithmic contribution, GRPO-N, is effectively identical to the previously proposed Negative Sample Reinforcement (NSR) method [1]. It follows the same underlying idea and exhibits the same empirical behavior. This undermines the claimed novelty of the submission.

[1] The Surprising Effectiveness of Negative Reinforcement in LLM Reasoning

**Questions:**

See weakness

---

> ### Author Response · Authors · 2025-11-14
> **Thank you!**
>
> Dear Reviewer xPjS,
>
> Thank you for taking the time to review our work and for offering thoughtful comments and constructive questions. Please see our detailed replies below. We have revised the manuscript and marked the changes in **orange**.
>
> > **W1**: The central “two-stage dynamics” claim lacks rigorous justification:
> >
> > ● The proposed two-phase interpretation remains largely speculative and is not formally or mathematically derived.
> >
> > ● The analysis does not identify when—or under what precise conditions—a transition between “shrinkage” and “expansion” should occur, nor does it guarantee that such a transition will happen in practice.
> >
> > ● In particular, if the model is already sufficiently exploitative, it may rarely sample low-confidence alternative solutions, making the proposed expansion phase unlikely and potentially leading instead to losing exploration ability.
>
> > **A**: That is a good question! We appreciate the opportunity to clarify this point.
> >
> > (1) For "rigorous justification", as Reviewer PFHy said that **"the proposed two-stage framework is inspiring and well supported by toy example and theoretical analysis"**. Also, Reviewer RBrT noted that **"Theoretically, it provides a mathematical formalization of this two-stage process"**. Therefore, we believe there may be some **misunderstandings** in your view that "the proposed two-phase interpretation remains largely speculative and is not formally or mathematically derived."
> >
> > (2) Regarding the "precise conditions," **as elaborated in Theorem 1 (Remark 3)**: (i) Shrinkage: Over-exploitation during the initial exploitation stage may lead to a shrinkage of the capability boundary. (ii) Expansion: As training progresses into the exploration stage, the growth rate of the previously dominant high-reward token decelerates as it approaches saturation (1 − π → 0). When a potentially optimal token—which now receives positive advantage signals—is occasionally sampled, its probability increases, while that of the formerly dominant high-reward token correspondingly declines.
> >
> > (3) **We strongly agree with your insight** that "if the model is already sufficiently exploitative, it may rarely sample low-confidence alternative solutions, making the proposed expansion phase unlikely and potentially leading instead to losing exploration ability." Indeed, **this is precisely why** we advocate avoiding over-exploitation during the Exploitation stage (the first stage in the two-stage dynamic), thereby facilitating a easy transition to the Exploration stage. **Guided by these findings, one can develop new algorithms to foster more advanced reasoning capabilities. The GSPO-N strategy serves as a straightforward implementation validating the guiding principles of our theory.**
>
> > **W2**: GRPO-N lacks novelty and is equivalent to prior NSR: The paper’s main algorithmic contribution, GRPO-N, is effectively identical to the previously proposed Negative Sample Reinforcement (NSR) method [1]. It follows the same underlying idea and exhibits the same empirical behavior. This undermines the claimed novelty of the submission.
> >
> > [1] The Surprising Effectiveness of Negative Reinforcement in LLM Reasoning
>
> > **A**: We sincerely thank the reviewer for raising this important point. We appreciate the opportunity to further summarize and clarify our work's contribution in relation to NSR [1]. See Section 6 in the revised version for details.
> >
> > (1) Differences in Experimental Setup：
> >
> > In NSR (their Appendix D.3) "we disable this normalization in PSR and NSR’s implementation" and "the advantage is equal to the raw reward (i.e., +1 for PSR and −1 for NSR)." Therefore, more precisely, the **NSR represents negative-reward gradients and may not account for relative changes during the training process**.
> >
> > In contrast, GRPO-N that uses **negative-advantage gradients** (A=r-mean(r)/std). As discussed in our theoretical analysis, a key feature of the **two-stage dynamic is the shift in relative negative samples**: from initially low-reward tokens (Figure 1 a3) to previously high-reward tokens (Figure 1 a1). Consequently, here, positive reward samples still serve the function of computing relative advantages by providing learning signals.
> >
> > (2) Theoretical contribution:
> > We are delighted that Reviewer RBrT have recognized the value of our derivations linking relative advantages to probability mass reallocation. As also noted by Reviewer PFHy, the **core contribution of this paper resides in the introduction of a two-stage framework to elucidate the dynamics of reasoning capabilities in RLVR. Our target is to establish a theoretical and empirical basis for understanding the impact of RLVR on reasoning capabilities**.
>
> The GRPO-N strategy serves only as a straightforward implementation validating the guiding principles of our theory. Additionally, Appendix C.5 provides a method for finer-grained control of probability assignments, inspired by our theoretical analysis.

---

> ### Author Response · Authors · 2025-11-27
> **Looking forward to your reply!**
>
> Dear Reviewer xPjS,
>
> We hope this message finds you well.
>
> We have carefully addressed the your comments and made substantial revisions to our manuscript, as detailed in the responses and the revised PDF file. We have also summarized the key changes and enriched content to facilitate your review.
>
> Given that the review timeline is approaching its deadline, **we kindly request your feedback on the revised submission at your earliest convenience**. Your insights and comments are crucial for further improving the quality of our work, and we greatly value the opportunity for continued discussion.
>
> Thank you very much for your time and effort. Please do not hesitate to let us know if there are any additional clarifications or further details needed.
>
> **Sending our best wishes for health, happiness, and all good things in life！**
>
> The Authors

---

### Official Review · Reviewer_rxbi · 2025-11-01

**Soundness:** 3
**Presentation:** 3
**Contribution:** 3
**Rating:** 4
**Confidence:** 3

**Summary:**

This paper addresses an ongoing controversy in reinforcement learning with verifiable rewards of whether RLVR shrinks or expands reasoning capability boundaries. The authors investigate probability mass dynamics, identifying two phases of training: An exploitation stage (associated with boundary shrinkage) and an exploration stage (associated with expansion). Based on their analysis, the paper proposes modified algorithms that use only relative negative gradients. Results show improved reasoning diversity and competitive Pass@k scores.

**Strengths:**

- The proposed two-stage dynamic provides an interesting and compelling conceptual synthesis: early exploitation leads to capability concentration, followed by later exploration leading to expansion.
- The paper is well motivated, written, and easy to follow.
- The experimental section is comprehensive, spanning multiple datasets with in- and out-of-domain evaluations.
- Experiments with Qwen2.5-Math-7B show improvements over the base model for large k.

**Weaknesses:**

- The theoretical analysis approximates importance ratios as ~1 and omits regularizers (KL, clipping). While this is justified for simplicity, it could overlook how they affect the two-stage transition in real training.
- While the authors' variant with Llama-3.2-3B-Instruct improves upon GRPO, the method still suffers from capability shrinkage, as indicated by the lower Pass@k metrics compared to the base model (Appendix C.3). The paper may benefit from more depth in the discussion of the comparison between Qwen and Llama.
- The experiments appear to run for fewer than 50 training steps (Fig. 2). Given that the paper’s central argument hinges on dynamics observable during prolonged training, this experimental horizon seems too short to observe the dynamics fully.

**Questions:**

- How do clipping and KL-regularization alter the two-stage dynamics?
- Why is the performance of Llama-3.2-3B-Instruct compared to the base model worse than the metrics of Qwen2.5-Math-7B?
- Why does GSPO-N lower test-entropy compared to GSPO?
- When using only negative gradients, what is the impact on training stability?
- Why did you only train for fewer than 50 training steps?

---

> ### Author Response · Authors · 2025-11-14
> **Thank you! (part 1)**
>
> Dear Reviewer rxbi,
>
> Thank you for your careful evaluation and thoughtful suggestions. We deeply value your time and effort in providing such constructive feedback. Below, we address your comments point by point. In response to all comments, we have revised the manuscript and marked the changes in **orange**.
>
> > **W1**: The theoretical analysis approximates importance ratios as ~1 and omits regularizers (KL, clipping). While this is justified for simplicity, it could overlook how they affect the two-stage transition in real training.
> >
> > **Q1**: How do clipping and KL-regularization alter the two-stage dynamics?
>
> > **A**: That is a good question! We appreciate the opportunity to clarify this point (**Section 6 in the revised version**).
> >
> > (1) For theoretical analysis. This simplification was made primarily to **highlight the core two-stage dynamics of policy optimization** under a simplified setting, allowing us to isolate and analyze the fundamental learning stages.
> >
> > (2) For real training. As we stated in Section 2, we acknowledge that in practical implementations, both clipping and KL regularization **play important roles in stabilizing training**. However, such **simplifications can be justified by empirical evidence**: prior work [1] has shown that omitting the clipping operation does not lead to performance degradation, while studies [1, 2] have demonstrated that the KL term can be removed when other hyperparameters are appropriately tuned.
> >
> > (3) For the effect of clipping and KL-regularization. [3] reports that  despite the reference KL achieves stable entropy values, it **fails to improve policy and instead leads to a degradation in performance**. [4] identifies that the **upper clip can restrict the exploration** of the policy, where making an ‘exploitation’ token more probable is much easier yet the probability of an unlikely ‘exploration’ token is too tightly bounded to be uplifted. Therefore, excessive KL-regularization and clipping will lead to limited exploration and premature policy determinization, thereby suppressing the emergence of the second phase in the two-stage dynamics.
> >
> > [1] Open-Reasoner-Zero: An Open Source Approach to Scaling Up Reinforcement Learning on the Base Model. NeurIPS 2025.
> >
> > [2] GPG: A Simple and Strong Reinforcement Learning Baseline for Model Reasoning. arXiv:2504.02546.
> >
> > [3] The Entropy Mechanism of Reinforcement Learning for Reasoning Language Models. arXiv:2505.22617.
> >
> > [4] DAPO: An Open-Source LLM Reinforcement Learning System at Scale. arXiv:2503.14476.
>
> > **W2**：While the authors' variant with Llama-3.2-3B-Instruct improves upon GRPO, the method still suffers from capability shrinkage, as indicated by the lower Pass@k metrics compared to the base model (Appendix C.3). The paper may benefit from more depth in the discussion of the comparison between Qwen and Llama.
> >
> > **Q2**: Why is the performance of Llama-3.2-3B-Instruct compared to the base model worse than the metrics of Qwen2.5-Math-7B?
>
> > **A**: Thank you for this insightful observation and suggestion. As we shown in Lines 429-430, "the performance ceiling is related to the base model, yet the key finding remains consistent across different models". We believe there are two primary factors contributing to this result:
> >
> > (1) Model Scale: The Qwen2.5-Math-7B model has significantly more parameters (7B vs. 3B), which generally provides a stronger foundational capacity for complex reasoning tasks like code generation. **The larger model is inherently more capable of retaining broad knowledge and skills during specialized training**.
> >
> > (2) **Base Model Specialization & Training Data**: The Qwen2.5-Math model was likely pre-trained and/or fine-tuned on a corpus with a stronger focus on mathematical reasoning and code. In contrast, the general-purpose Llama-3.2-3B base model might not have the same inherent bias towards these specific tasks. Consequently, the instruction-tuning or alignment process for Llama might lead to a more noticeable trade-off, where improving alignment on a specific distribution can sometimes come at the cost of broader capabilities, especially in a smaller model.
> >
> > We agree that a deeper comparative discussion between the Qwen and Llama families would strengthen the paper. In the revised version, **we expand on this point in the Appendix C.3**.
>
> We firmly believe that **a good theory should not only account for empirical observations but also provide actionable insights for design**.
>
> The GRPO-N strategy serves as a straightforward implementation validating the guiding principles of our theory. Additionally, Appendix C.5 provides a method for finer-grained control of probability assignments, inspired by our theoretical analysis.

---

> > ### Author Response · Authors · 2025-11-14
> > **Thank you! (part 2)**
> >
> > > **W3**: The experiments appear to run for fewer than 50 training steps (Fig. 2). Given that the paper’s central argument hinges on dynamics observable during prolonged training, this experimental horizon seems too short to observe the dynamics fully.
> > >
> > > **Q5**:  Why did you only train for fewer than 50 training steps?
> >
> > > **A**: We sincerely appreciate your thorough comment. We are glad to have the opportunity to clarify.
> > >
> > > (1) As stated in Section 4.2, "For the datasets, we employ the training set of MATH, which comprises 7,500 problems, for model training (with a prompt batch size of 1,024)." Therefore, approximately 7 training steps are sufficient to complete one full pass through the training dataset. Additionally, as shown in Figure 2, the model begins to converge after just over ten steps. Hence, we believe that using fewer than 50 training steps is reasonable for this experimental setup. (2) We also acknowledge that longer training durations would better capture the full dynamics of the process. As noted in Lines 387–390, prior study [1] suggests that policy performance improves at the cost of policy entropy and is therefore limited by its exhaustion. Thus, the model optimized by GRPO‑N can serve as a strong baseline while preserving the base model’s diversity, making it suitable for prolonged training.
> > >
> > > [1] The Entropy Mechanism of Reinforcement Learning for Reasoning Language Models. arXiv:2505.22617.
> >
> > > **Q3**: Why does GSPO-N lower test-entropy compared to GSPO?
> > >
> > > **A**: This is a pertinent question. We add this discussion in the revised version (Section 6).
> > >
> > > (1) First, we recall the key distinction between GSPO and GRPO, as outlined in Lines 138–142: while the token-level importance ratio $w_{i,t}$ in GRPO may not align well with sequence-level rewards, GSPO introduces a sequence-level importance ratio $w_i$​ based on sequence likelihood.
> > >
> > > (2) Moreover, certain tokens (such as “wait”) that frequently appear in both positive and negative rollouts can lead to instability when optimized with token-level importance ratios. As illustrated in Figure 2, this explains why both GSPO and GSPO-N exhibit greater stability compared to GRPO and GRPO-N.
> > >
> > > (3) Regarding the difference between GSPO-N and GSPO: as noted in Appendix C.2, we use a temperature of 0.6 and a top‑p value of 0.95 during testing. We believe one reason for the observed difference is the stochasticity inherent in reinforcement learning, such as that introduced by online sampling. On the other hand, although GSPO-N and GSPO use the same learning rate, GSPO-N does not explicitly reinforce positive rollouts, which may result in slightly slower learning. However, as shown in Figure 2, the performance gap between the two is not substantial.
> >
> > > **Q4**: When using only negative gradients, what is the impact on training stability?
> > >
> > > **A**: (1) **Theoretically, as discussed in Remark 2 (Lemma 1) and Remark 3 (Theorem 1)**, when the advantage estimate is negative, i.e., $Aˆ(v) < 0$, the policy update increases the probabilities of other actions proportionally to their current policy values. This mechanism allows the update to reallocate probability mass toward other potentially advantageous actions that may have been previously under-sampled. Consequently, using only negative gradients helps mitigate over-exploitation during the exploitation phase, thereby enhancing training stability by promoting a more balanced exploration of the action space.
> > >
> > > (2) **Empirically**, as shown in Figures 2 and 4, using only relative negative gradients (GRPO-N) help mitigate overconfidence in previously sampled responses. The model optimized by GRPO-N may be a good baseline and maintain the base model’s diversity for prolonging training. Figure 3 shows that GRPO assigns higher probability to the entire trajectory (i.e., the behavior of generating incorrect code is reinforced—a tendency that may significantly affect the model’s self-correction ability), whereas GRPO-N mitigates this issue.

---

> ### Author Response · Authors · 2025-11-27
> **Looking forward to your reply!**
>
> Dear Reviewer rxbi,
>
> We hope this message finds you well.
>
> We have carefully addressed the your comments and made substantial revisions to our manuscript, as detailed in the responses and the revised PDF file. We have also summarized the key changes and enriched content to facilitate your review.
>
> Given that the review timeline is approaching its deadline, **we kindly request your feedback on the revised submission at your earliest convenience**. Your insights and comments are crucial for further improving the quality of our work, and we greatly value the opportunity for continued discussion.
>
> Thank you very much for your time and effort. Please do not hesitate to let us know if there are any additional clarifications or further details needed.
>
> **Sending our best wishes for health, happiness, and all good things in life！**
>
> The Authors

---

### Official Review · Reviewer_RBrT · 2025-11-02

**Soundness:** 3
**Presentation:** 3
**Contribution:** 3
**Rating:** 6
**Confidence:** 3

**Summary:**

The paper addresses the ongoing discussion on RLVR by making the proposition that RLVR can both shrink and expand LLM reasoning capability, in two phases. Early exploitation stage sharpens distribution and narrows support; the later exploration stage redistributes mass to newly sampled, high-reward regions.

Theoretically, it provides a mathematical formalization of this two-stage process. Methodologically, based on this analysis, the paper proposes GRPO-N and GSPO-N, variants of group policy optimization algorithms. These methods operate by using "exclusively relative negative gradients".1 The authors claim this approach mitigates entropy collapse and enables the stable training required to reach exploration stage.

**Strengths:**

- The paper addresses a timely topic, so it is very interesting to read this paper.

- The theoretical contribution us simple derivations that connect relative advantages to probability mass reallocation.

- The empirical observation on entropy collapse and rebound aligns with other findings that RLVR often collapses entropy early without intervention.

**Weaknesses:**

- The underlying problem of understanding RLVR dynamics is both valid and important. However, the paper’s framing of it as an unresolved paradox (“shrinkage, expansion, or both?”) is not well supported. Cui et al. clearly show that entropy drops sharply during the early training stage, with most performance gains and the associated entropy collapse occurring early on (see figure 2 in their work). Liu et al. demonstrate that “sustained gains” require extended training, using stability mechanisms such as policy resets to expand reasoning boundaries. The literature does not present a real debate or contradiction—rather, it reflects observations of different phenomena occurring at different time scales.


- The paper's core method, GRPO-N that uses only negative-advantage gradients appears to be a direct rediscovery of NSR (fZhu et al.) The authors cite this work but fail to compare against it or differentiate their contribution.

**Questions:**

- The paper claims that GRPO-N's entropy "significantly surpasses that of the base model". Given GRPO-N only punishes bad actions and never reinforces good ones, isn't this "high entropy" simply the policy flattening by learning what not to do, without ever converging on what to do, as explained by Lemma 1? How do you explain this "high entropy" with GRPO-N's Pass@1 performance in Table 1?

- GRPO-N method discards all positive-gradient samples. This seems sample-inefficient. Have you measured the wall-clock time and number of samples GRPO-N requires to converge, compared to standard GRPO?

---

> ### Author Response · Authors · 2025-11-14
> **Thank you! (part 1)**
>
> Dear Reviewer RBrT,
>
> We sincerely appreciate your thorough review and the insightful comments you provided. Your feedback is invaluable in improving our paper. In response to all comments, we have revised the manuscript and marked the changes in **orange**. Please find our detailed response below.
>
> > **W1**: The underlying problem of understanding RLVR dynamics is both valid and important. However, the paper’s framing of it as an unresolved paradox (“shrinkage, expansion, or both?”) is not well supported. Cui et al. clearly show that entropy drops sharply during the early training stage, with most performance gains and the associated entropy collapse occurring early on (see figure 2 in their work). Liu et al. demonstrate that “sustained gains” require extended training, using stability mechanisms such as policy resets to expand reasoning boundaries. The literature does not present a real debate or contradiction—rather, it reflects observations of different phenomena occurring at different time scales.
>
> > **A**: Thank you for this critical observation regarding the temporal progression of RLVR training. Our central thesis is, in fact, that these phenomena are stage-dependent, and we aimed to synthesize the **contrasting evidence (please see Section 1 for details)** into a coherent discussion of a two-stage dynamic process.  **We adopt a more precise expression in the revised version, such as "is still not fully resolved"**.
> >
> > We also wish to **gently clarify a potential point of misunderstanding**:
> >
> > (1) We agree with your observation that "Cui et al. [1] clearly show that entropy drops sharply during the early training stage...". However, it is crucial to distinguish that performance gains are not equivalent to capability boundary expansion or shrinkage, as established in our theoretical analysis (see Lemma 1 and Theorem 1). Furthermore, Figure 2 of their work demonstrates that entropy exhibits a consistent downward trend throughout the observed training period. Besides, [2] indicates limited exploration and early deterministic policy, which can hinder the scaling process.
> >
> > (2) As rightly noted by **Reviewer PFHy**, the main contribution of this paper lies in introducing the two-stage framework to interpret the reasoning capability dynamics in RLVR. Our primary goal is to leverage this framework to systematically explain the observed divergent phenomena, with the ultimate goal of guiding more effective algorithm design.
> >
> > [1] The Entropy Mechanism of Reinforcement Learning for Reasoning Language Models. arXiv:2505.22617.
> >
> > [2] DAPO: An Open-Source LLM Reinforcement Learning System at Scale. arXiv:2503.14476.
>
> > **W2**: The paper's core method, GRPO-N that uses only negative-advantage gradients appears to be a direct rediscovery of NSR (Zhu et al.) The authors cite this work but fail to compare against it or differentiate their contribution.
>
> > **A**: We sincerely thank the reviewer for raising this important point. We appreciate the opportunity to further summarize and clarify our work's contribution in relation to NSR [1]. **See Section 6 in the revised version**.
> >
> > (1) **Differences in Experimental Setup**：
> >
> > In NSR (their Appendix D.3) "we disable this normalization in PSR and NSR’s implementation" and "the advantage is equal to the raw reward (i.e., +1 for PSR and −1 for NSR)." Therefore, more precisely, the NSR represents **negative-reward gradients** and may not account for relative changes during the training process.
> >
> > In contrast, GRPO-N uses **negative-advantage gradients** (A=r-mean(r)/std). As discussed in our theoretical analysis, a key feature of the two-stage dynamic is the shift in relative negative samples: from initially low-reward tokens (Figure 1 a3) to previously high-reward tokens (Figure 1 a1). Consequently, here, positive reward samples still serve the function of computing relative advantages by providing learning signals.
> >
> > (2) **Theoretical contribution**:
> >
> > We are delighted that you have recognized the value of our derivations linking relative advantages to probability mass reallocation. As also noted by Reviewer PFHy, **the core contribution of this paper resides in the introduction of a two-stage framework to elucidate the dynamics of reasoning capabilities in RLVR**. Our target is to establish a theoretical and empirical basis for understanding the impact of RLVR on reasoning capabilities.
> >
> > [1] The Surprising Effectiveness of Negative Reinforcement in LLM Reasoning. NeurIPS 2025.
>
> We firmly believe that **a good theory should not only account for empirical observations but also provide actionable insights for design**.
>
> The GRPO-N strategy serves as a straightforward implementation validating the guiding principles of our theory. Additionally, Appendix C.5 provides a method for finer-grained control of probability assignments, inspired by our theoretical analysis.

---

> > ### Author Response · Authors · 2025-11-14
> > **Thank you! (part 2)**
> >
> > > **Q1**: The paper claims that GRPO-N's entropy "significantly surpasses that of the base model". Given GRPO-N only punishes bad actions and never reinforces good ones, isn't this "high entropy" simply the policy flattening by learning what not to do, without ever converging on what to do, as explained by Lemma 1? How do you explain this "high entropy" with GRPO-N's Pass@1 performance in Table 1?
> >
> > > **A**: We thank the reviewer for this excellent and thoughtful question. It gets to the heart of a key mechanism in our analysis.
> > >
> > > Your statement that "GRPO-N only punishes bad actions and never reinforces good ones" **is not entirely accurate**. GRPO-N actually penalizes actions with low relative advantages, and as the model progresses through learning stages, the set of actions considered to have low relative advantages changes under online sampling (**A to W2**). Therefore, the "high entropy" reflects the preservation of greater diversity in the model's generative space rather than a tendency toward flattening. The model still converges to correct behaviors, as demonstrated in Table 1 and Figure 2. The "high entropy" observed in GRPO-N corresponds to an improvement in Pass@1 performance alongside the preservation of greater diversity.
> >
> > > **Q2**: GRPO-N method discards all positive-gradient samples. This seems sample-inefficient. Have you measured the wall-clock time and number of samples GRPO-N requires to converge, compared to standard GRPO?
> >
> > > **A**: That is a good question! As discussed above, GRPO-N does **not completely discard** positive-gradient samples. We still utilize them to compute relative advantages for policy updates, but simply mask these positive-advantage gradients during backpropagation.
> > >
> > >  For "wall-clock time and number of samples", as shown in Figures 2 and 3, despite comparable requirements in wall-clock time and number of samples for convergence, GRPO-N preserves significantly higher entropy, thereby retaining greater exploratory capacity.
> > >
> > >  In addition, **you have raised a constructive direction for future work**. We could indeed improve sample efficiency by implementing a prudent selection process for these positive samples, building upon our initial analysis of different correct responses in Section 4.3.

---

> ### Author Response · Authors · 2025-11-27
> **Looking forward to your reply!**
>
> Dear Reviewer RBrT,
>
> We hope this message finds you well.
>
> We have carefully addressed the your comments and made substantial revisions to our manuscript, as detailed in the responses and the revised PDF file. We have also summarized the key changes and enriched content to facilitate your review.
>
> Given that the review timeline is approaching its deadline, **we kindly request your feedback on the revised submission at your earliest convenience**. Your insights and comments are crucial for further improving the quality of our work, and we greatly value the opportunity for continued discussion.
>
> Thank you very much for your time and effort. Please do not hesitate to let us know if there are any additional clarifications or further details needed.
>
> **Sending our best wishes for health, happiness, and all good things in life！**
>
> The Authors

---

### Author Response · Authors · 2025-12-01
**General Response (part 1)**

Dear Area Chair, Senior Area Chair and Program Chair,

We would like to express our deepest gratitude for your generous time and patience.

While our point-by-point rebuttal has directed the reviewers to the exact locations, **we wish to provide a concise summary of these responses for your convenience**, given that they are cardinal to the evaluation of our work's contribution and completeness.

We are grateful to all reviewers for their detailed and constructive feedback! We are encouraged to see that reviewers find (``novelty, theoretical and empirical contribution, good writing...``):

> ``Reviewer RBrT``: The paper **addresses a timely topic**, so it is very **interesting to read** this paper...... The **theoretical contribution** us simple derivations that **connect relative advantages to probability mass reallocation**.
>
> ---
>
> ``Reviewer rxbi``: The proposed **two-stage dynamic provides an interesting and compelling conceptual** synthesis: early exploitation leads to capability concentration, followed by later exploration leading to expansion...... The paper is **well motivated, written, and easy to follow**...... The **experimental section is comprehensive**......
>
> ---
> ``Reviewer xPjS``: **Discussing a real and important empirical debate** in RLVR reasoning...... The toy **helps intuition**...... **Experiments cover multiple** baselines and several reasoning benchmarks......
>
> ---
> ``Reviewer PFHy``: **Timely and relevant** topic...... The paper addresses it with **both theoretical and empirical clarity**.......  Elegant **theoretical framing and well-designed** toy example The derivation of Lemma 1 + Theorem 1 connects token-level probability dynamics to capability boundary movement, **presenting a physically interpretable “force-like” view similar to learning-dynamics literature**. The intuition that action a1 saturates and then the confidence of action a2 gradually increases is **very novel and inspiring**.
>
> (Reviewer PFHy's **``final reply (long before the ICLR-reported "bug", can be checked in the Revisions)``** https://openreview.net/revisions?id=0hS79AdLYE: "the authors have made substantial efforts to clarify motivation, structure, and algorithmic implications. Taking these improvements into account, I am inclined to **``raise my score`` from 4 to 6."**)

We have addressed all the questions raised by the reviewers through detailed clarifications, providing separate responses to each reviewer. The revisions are marked with ``orange`` text in the PDF. We summarize these points below.

> **P1: The main contribution of this work**.
>
> We firmly believe that a good theory should not only account for empirical observations but also provide actionable insights for design.
>
> As noted by ``Reviewer PFHy``, "**the main contribution of this paper lies in introducing the two-stage framework to interpret the dynamics of reasoning capability in RLVR, rather than proposing a new algorithm**". Our target is to establish a theoretical and empirical basis for understanding the impact of RLVR on reasoning capabilities. **Guided by these findings, one can develop new algorithms to foster more advanced reasoning capabilities**. The GRPO-N strategy serves as a straightforward implementation **validating the guiding principles of our theory**.  Additionally, Appendix C.5 provides another method GRPO-TN for finer-grained control of probability assignments, inspired by our theoretical analysis.

> **P2: Differences from prior work**, ``for Reviewers RBrT, xPjS, PFHy``:  Negative Reward Gradients \& Negative Advantage Gradients.
>
> **Beyond the theoretical components**, we further clarify the methodological and conceptual differences between our work and NSR [1].
>
> (1) As documented in Appendix D.3 of the NSR paper, normalization is disabled in their implementation, and the advantage is defined as the raw reward (i.e., +1 for PSR and -1 for NSR). Consequently, NSR is more precisely described as method using **negative reward gradients**, which does not account for relative changes over the course of training.
>
> (2) In contrast, GRPO-N uses **negative advantage gradients** ($r - mean(r) / std(r)$). **As discussed in our theoretical analysis (Thereom 1), a key feature of the two-stage dynamic is the shift in relative negative samples: from initially low-reward tokens (Figure 1 $a_3$) to previously high-reward tokens (Figure 1 $a_1$)**. Consequently, here, positive reward samples still serve the function of computing relative advantages by providing learning signals.
>
> [1] The Surprising Effectiveness of Negative Reinforcement in LLM Reasoning. arXiv:2506.01347.

---

> ### Author Response · Authors · 2025-12-01
> **General Response (part 2)**
>
> > **P3: For the the validity of the assumptions**.
> >
> > - Rationale for Omitting Regularization (KL, Clipping), for ``Reviewer rxbi``:
> >
> > (1) For theoretical analysis. This simplification was made primarily to **highlight the core two-stage dynamics of policy optimization** under a simplified setting, allowing us to isolate and analyze the fundamental learning stages.
> >
> > (2) For real training. As we stated in Section 2, we acknowledge that in practical implementations, both clipping and KL regularization **play important roles in stabilizing training**. However, such **simplifications can be justified by empirical evidence**: prior work [1] has shown that omitting the clipping operation does not lead to performance degradation, while studies [1, 2] have demonstrated that the KL term can be removed when other hyperparameters are appropriately tuned.
> >
> > (3) For the effect of clipping and KL-regularization. [3] reports that  despite the reference KL achieves stable entropy values, it **fails to improve policy and instead leads to a degradation in performance**. [4] identifies that the **upper clip can restrict the exploration** of the policy, where making an ‘exploitation’ token more probable is much easier yet the probability of an unlikely ‘exploration’ token is too tightly bounded to be uplifted. Therefore, excessive KL-regularization and clipping will lead to limited exploration and premature policy determinization, thereby suppressing the emergence of the second phase in the two-stage dynamics.
> >
> > [1] Open-Reasoner-Zero: An Open Source Approach to Scaling Up Reinforcement Learning on the Base Model. NeurIPS 2025.
> >
> > [2] GPG: A Simple and Strong Reinforcement Learning Baseline for Model Reasoning. arXiv:2504.02546.
> >
> > [3] The Entropy Mechanism of Reinforcement Learning for Reasoning Language Models. arXiv:2505.22617.
> >
> > [4] DAPO: An Open-Source LLM Reinforcement Learning System at Scale. arXiv:2503.14476.
> >
> > ---
> >
> > - A key assumption in the above reasoning is that r1<r2, for ``Reviewer PFHy``.
> >
> > As we described in Section 2, in most practical implementations of GRPO/GRPO-N, the coefficient $w_{i,t}\hat{A}_{i,t}$ is various among different correct responses. NSR uses only the raw-reward $r_i$. (see **P2** for more details).
> >
> > Specifically, the gradient in the actual RLVR setup is proportional to $w_{i,t}\hat{A}_{i,t}\nabla \log \pi$, rather than $r_i\nabla \log \pi$. On this basis, the assumption is reasonable, and we are confident that the theoretical and experimental components of our paper are aligned and self-contained.
> >
> > ---
> >
> > For whether the two-stage mechanism rigorously holds under the 1/0 reward structure, the answer depends on the specific case:
> >
> > **One the one hand**, If we consider the entire sentence as a complete action sample, and if both actions a1 and a2 receive a reward of 1 (i.e., both are “good” actions), then—as analyzed in Remarks 2 and 3—the update depends on both the advantage estimate A and the current policy distribution π. Under the two-stage dynamic analysis, neither p(a1) nor p(a2) will converge to 0 in the limit. **On the other hand**, if we model each token as an action, a sentence is generated through a sequence of multiple action samplings. Under this perspective, the sampling space expands to $O(V^T)$, as outlined in Section 3. Here, the $w_{i,t}\hat{A}_{i,t}$ for every token is various, the two-stage mechanism still holds, this may be  "under non-binary reward settings to validate the assumptions".
>
> > **P4:** Regarding the "precise conditions" in two-stage dynamics (for ``Reviewers xPjS``).
> >
> > **As elaborated in Theorem 1 (Remark 3)**: (i) Shrinkage: Over-exploitation during the initial exploitation stage may lead to a shrinkage of the capability boundary. (ii) Expansion: As training progresses into the exploration stage, the growth rate of the previously dominant high-reward token decelerates as it approaches saturation (1 − π → 0). When a potentially optimal token—which now receives positive advantage signals—is occasionally sampled, its probability increases, while that of the formerly dominant high-reward token correspondingly declines. (see A to Reviewers xPjS'W1 for more details.)
>
> Thank you again for your time and consideration. Many of us spent a significant amount of work on the rebuttal and the subsequent ensuing discussion. **We sincerely hope that honest authors receive fair recognition**, and we trust that our general response will be helpful for you.
>
> **Sending our best wishes for health, happiness, and all good things in life!**
>
> The Authors.

---

### Meta-Review · Area_Chair_Q6CD · 2025-12-26

**Summary:**

After reading the reviews, rebuttal, and the paper, I find the submission below the bar for a top-tier venue. The central claim—an inherent “two-stage” RLVR dynamic in which capability boundaries first shrink and later expand—is not solidified as a formal theoretical statement, but instead built from a mix of elementary derivative identities and interpretive narrative. This concern is shared with Reviewer xPjS.

Concretely, the technical core (Lemma 1 / Theorem 1) largely repackages standard softmax/REINFORCE-style gradient structure into an expectations expression . The “two-stage” conclusion is then derived primarily through interpretation (Remark 3), rather than a rigorous mechanism with clearly stated conditions. Consequently, the work reads as speculative: it offers a plausible story consistent with known gradients, but does not rigorously justify that the proposed transition should happen in practice, especially in regimes where policies become exploitative and rarely sample low-confidence alternatives.

I recommend rejection.

**Reviewer Concerns:**

Addressed:
* why KL/clipping can be "ignored" in theory (I am also ok with this simplifying assumtpion for "theory")
* why their GRPO-N differs in motivation from same algorithm in previous work and better comparing with that

Outstanding:
* Triviality / presentation of “theory”: Lemma 1 is essentially the textbook gradient of log-softmax leading to the familiar “increase sampled logit, decrease others” form; calling this a lemma overstates its contribution
* Lack of rigorous justification for the two-stage dynamics: Theorem 1 gives an expected logit update, but the claimed “exploitation → exploration” staging is asserted as a "Remark"  without a formal derivation, but later (also in the discussion refered to as "theory")

**Reviewer Scores:**

RbRT: 6-->6
rxbi: 4-->4
xPjS: 4-->4
PFHy: 4---> potentially raise to 6 with hesitation or maintained their 4

---

### Decision · Program_Chairs · 2026-01-26

Reject